

# A missing link in the carbon cycle: phytoplankton light absorption under RCP scenarios

Rémy Asselot[1,*], Frank Lunkeit[2], Philip Holden[3], and Inga Hense[1]

[1]Institute for Marine Ecosystem and Fishery Science, Center for Earth System Research and Sustainability, University of Hamburg, Hamburg, Germany
[2]Meteorological Institute, Center for Earth System Research and Sustainability, University of Hamburg, Hamburg, Germany
[3]Environment, Earth and Ecosystems, The Open University, Walton Hall, Milton Keynes, MK7 6AA, UK
[*]Now at University of Brest, CNRS, Ifremer, IRD, Laboratoire d'Océanographie Physique et Spatiale (LOPS), IUEM, F29280, Plouzané, France.

**Correspondence:** Rémy Asselot (remy.asselot@ifremer.fr)

**Abstract.** Marine biota and biogeophysical mechanisms, such as phytoplankton light absorption, have attracted increasing attention in recent climate studies. Under global warming, the impact of phytoplankton on the climate system is expected to change. Previous studies analyzed the impact of phytoplankton light absorption under prescribed future atmospheric $CO_2$ concentrations. However, the role of this biogeophysical mechanism under freely-evolving atmospheric $CO_2$ concentration and future $CO_2$ emissions remains unknown. To shed light on this research gap, we perform simulations with the EcoGEnIE Earth system model and prescribe $CO_2$ emissions out to 2500 following the four Extended Concentration Pathways (ECP) scenarios, which for practical purpose we call RCP scenarios. Under all RCP scenarios, our results indicate that phytopankton light absorption weakens the biological carbon pump while it increases the surface chlorophyll, the sea surface temperature, the atmospheric $CO_2$ concentrations and the atmospheric temperature. Under the RCP2.6, RCP4.5 and RCP6.0 scenarios, the magnitude of changes due to phytoplankton light absorption is similar. However, under the RCP8.5 scenario, the changes in the climate system are less pronounced due to temperature limitation of phytoplankton concentration, highlighting a reduced effect of phytoplankton light absorption under strong warming. Additionally, this work highlights the major role of phytoplankton light absorption on the climate system, suggesting highly uncertain feedbacks on the carbon cycle with uncertainties that maybe in the range of those known from the land biota.

## 1 Introduction

With global warming, phytoplankton abundance and distribution are predicted to change but how these changes affect biogeophysical mechanisms such as phytoplankton light absorption remains unknown. Using an Earth system model (ESM) of intermediate complexity, we study the effect of phytoplankton light absorption on the climate system under future emission scenarios.

Under global warming, the future changes of phytoplankton biomass and primary production are highly uncertain. Observa-





tions indicate that the abundance of phytoplankton biomass has decreased due to global warming. For instance, oceanographic measurements from 1890 to 2010 reveal that chlorophyll concentration has declined over more than 62% of the ocean surface (Boyce et al., 2014). Additionally, Polovina et al. (2008) indicate that between 1998 and 2006, low surface chlorophyll areas

have expanded by 15% on a global scale although their results might not be exclusively attributed to climate change due to their short time series (Henson et al., 2010; Schlunegger et al., 2020). Using an ocean-color database spanning 6 years, Mc-Clain et al. (2004) show that the oligotrophic waters expand in the Northern hemisphere while the expansion in the Southern hemisphere is much weaker. Complementing these observations, modeling studies have also investigated the future changes in net primary production due to anthropogenic warming. For instance, a CMIP6 model-ensemble study indicates a decrease in

depth-integrated primary production of 2.99±9.11% by the end of the 21st century under the high emission scenario SSP5-8.5 (Kwiatkowski et al., 2020). However, this estimate is rather imprecise due to incomplete understanding and insufficient observational constraints; thus the projections of primary production changes show large uncertainties (Tagliabue et al., 2021). Furthermore, using a coupled ocean-biogeochemistry model, Couespel et al. (2021) highlight a decrease in net primary production of 12% after a linear increase in atmospheric temperature reaching +2.8°C by the end of the 21st century. These changes

in phytoplankton abundance, distribution and biogeography have consequently an impact on the role of phytoplankton light absorption.

Different modeling studies investigate the effect of phytoplankton light absorption under global warming. It is suggested that the decrease in phytoplankton abundance will increase ocean clarity and lead to a lower biological increase of sea surface tem-

perature (SST). A reduction of phytoplankton-induced oceanic warming could thus counteract in part the warming associated with climate change (Patara et al., 2012). To study the effect of phytoplankton light absorption in a warming scenario, Sonntag (2013) modified the oceanic forcing by increasing the sea surface temperature for the whole model domain by 3°C. Taking into account phytoplankton light absorption, surface phytoplankton concentrations are enhanced and the maximum SST increase is 0.4°C compared to a present-day scenario (Sonntag, 2013). Furthermore, Paulsen (2018) uses an Earth system model of

high complexity to perform simulations under a transient increase of 1% of atmospheric $CO_2$ per year. With phytoplankton light absorption, Paulsen (2018) reports a decline in chlorophyll concentrations and an enhanced circulation in the upwelling regions, leading to a local oceanic warming of up to 0.7°C. Following RCP8.5 scenario, Kvale and Meissner (2017) investigate the sensitivity of the light attenuation coefficient for phytoplankton. Depending of the parameterization choice, the authors highlight that phytoplankton light absorption may reduce or increase net primary production between 1800 and 2100. Addi-

tionally, using a coupled ocean-atmosphere model, Park et al. (2015) focus on the Arctic region to study phytoplankton light absorption under global warming. They conduct simulations where atmospheric $CO_2$ concentration increases by 1% per year from the level of 1990 to double its initial concentration. The authors show that phyoplankton light absorption amplifies future Arctic warming by 20%.

To date, the impact of phytoplankton light absorption under oceanic warming (Sonntag, 2013), constant atmospheric $CO_2$ concentration (Patara et al., 2012) and prescribed rising atmospheric $CO_2$ concentrations (Park et al., 2015; Kvale and Meiss-



ner, 2017; Paulsen, 2018) has been investigated. However, using an Earth System model of intermediate complexity, Asselot et al. (2022) study how atmospheric temperature is affected by phytoplankton light absorption. To do so, the authors compare the changes in air-sea heat versus air-sea $CO_2$ exchange due to this biogeophysical mechanism. They conclude that phyto-

plankton light absorption mainly affects the climate system via air-sea $CO_2$ exchange. Therefore, prescribing atmospheric $CO_2$ concentrations for global warming simulations blurs the real effect of this biogeophysical mechanism. As a consequence, rather than prescribing the atmospheric $CO_2$ concentrations, we are interested in the effects of phytoplankton light absorption under future $CO_2$ emissions on a long timescale. To address this question we apply the EcoGEnIE Earth system model (Ward et al., 2018) and force the atmosphere with $CO_2$ emissions following the four Representative Concentration Pathways (RCP)

scenarios used by the Intergovernmental Panel on Climate Change (IPCC) for their Fifth Assessment Report (Moss et al., 2010).

## 2   The Representative Concentration Pathways scenarios

The RCP scenarios describe possible future climate systems adopted by the IPCC (Moss et al., 2010) depending on the volume of greenhouse gases emitted in the next years (Figure 1). Originally, there were four RCP scenarios, namely RCP2.6, RCP4.5,

RCP6.0 and RCP8.5, labeled after a net enhancement of radiative forcing at the beginning of the $22^{nd}$ century (2.6, 4.5, 6.0 and 8.5 W/m$^2$, respectively). These scenarios are consistent with socio-economic assumptions and associated greenhouse gas emissions. They comprise a stringent mitigation scenario (RCP2.6), two intermediate scenarios (RCP4.5 and RCP6.0) and a high greenhouse gas emissions scenario (RCP8.5). The RCP scenarios only span the 2005-2100 period but this study is conducted on a multi-century timescale to understand the long term climate response. As a consequence, our study requires

data beyond 2100. We therefore use the Extended Concentration Pathways (ECPs) designed by stakeholders and scientific groups and spanning the 2100-2500 period (Meinshausen et al., 2011). Similar to RCP2.6, the ECP2.6 represents a strong mitigation scenario including negative $CO_2$ emissions from 2100 to 2500. For the ECP4.5 and ECP6.0, the atmospheric $CO_2$ emissions start to decrease in the $21^{st}$ century while for ECP8.5 this decrease happens at the end of the $22^{nd}$ century. For practical purposes, here, referring to the RCP scenarios indicate the period between 1765 and 2500.



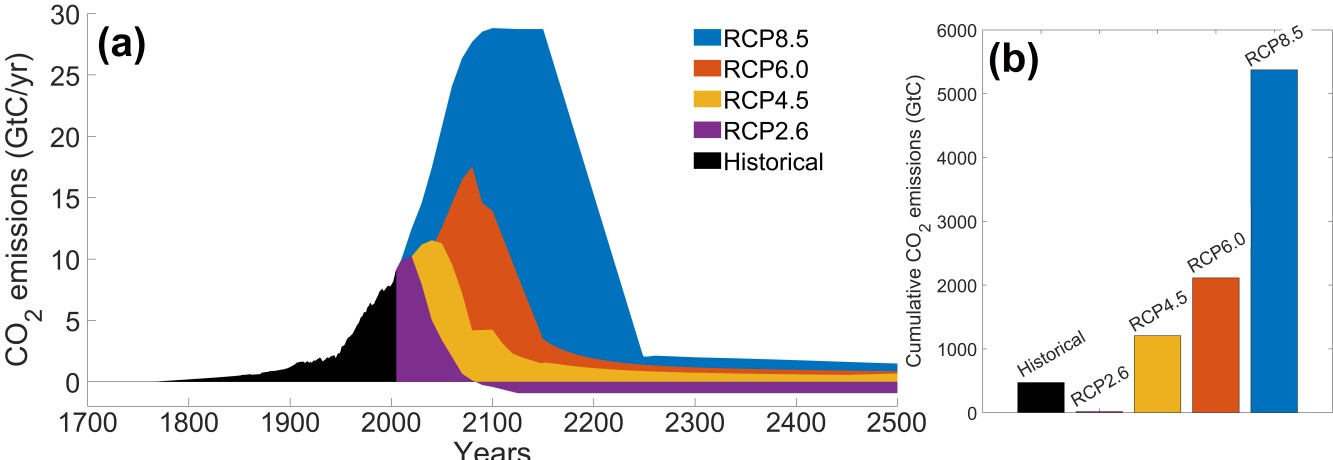

**Figure 1.** Atmospheric $CO_2$ emissions following the RCP scenarios. (a) Historical and predicted $CO_2$ emissions over time (GtC/yr). (b) Cumulative $CO_2$ emissions for the different scenarios (GtC). The historical emissions represent the cumulative $CO_2$ emissions from 1765 to 2005. The RCP scenarios represent the cumulative $CO_2$ emissions between 2006 and 2500. The color coding between the two panels is identical.

## 3 Methods

The ESM used in this study is called EcoGEnIE (Ward et al., 2018) and is an association between a new ecosystem component (ECOGEM) and a previous model named cGEnIE (Lenton et al., 2007). EcoGEnIE is an ESM of intermediate complexity (EMIC) (Claussen et al., 2002) and due to the limitation of such a model, we focus on the qualitative assessments rather than on quantitative estimates of our results. Moreover, cGEnIE is widely used to study past climate systems and the carbon cycle over geological timescales (Gibbs et al., 2016; Meyer et al., 2016; Greene et al., 2019; Stockey et al., 2021). EcoGEnIE was already used to analyze the role of marine phytoplankton in the warm early Eocene period (Wilson et al., 2018) and to explore the relationships between plankton size, trophic complexity and the availability of phosphorus during the late Cryogenian (Reinhard et al., 2020). We use the same configuration as described in Asselot et al. (2021). This model contains components related to climate processes, including ocean dynamics, marine biogeochemistry, marine ecosystem, atmospheric circulation and sea-ice dynamics (Figure 2). We do not consider a dynamical land scheme, thus the surface land temperature is equal to the surface atmospheric temperature. For this study, we modify the ecosystem component and the oceanic component to implement phytoplankton light absorption.





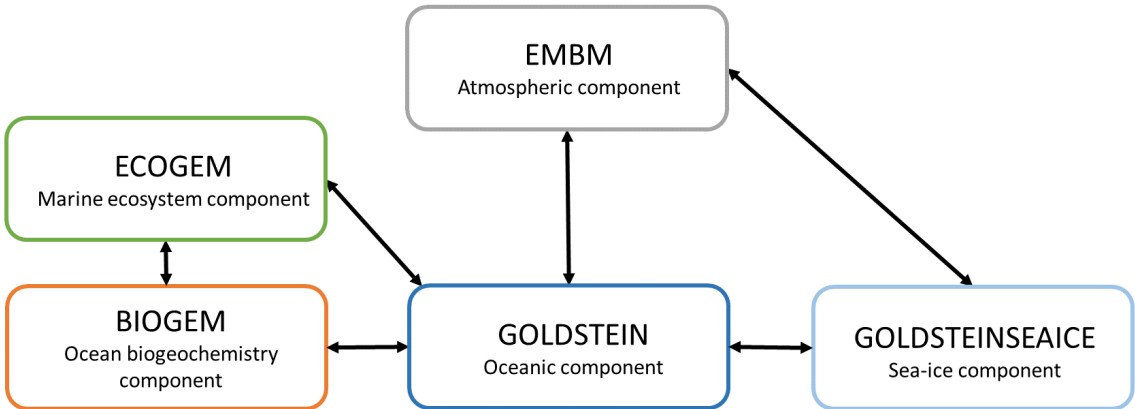

**Figure 2.** Sketch representing the different components of the EcoGEnIE model. Black arrows represent the links between the different components. Figure from Asselot et al. (2022).

## 3.1 Ocean, atmosphere and sea-ice representation

The oceanic component is a 3D frictional-geostrophic oceanic component (GOLDSTEIN) that calculates the horizontal and
vertical redistribution of heat, salinity and biogeochemical elements (Edwards and Marsh, 2005). The horizontal grid ($36 \times 36$) is uniform in longitude and uniform in sine latitude, giving $\sim 3.2°$ latitudinal increments at the equator increasing to $19.2°$ in the polar regions. This horizontal grid has been employed as the standard resolution to study the global carbon cycle (Cameron et al., 2005). Furthermore, we consider 32 vertical oceanic layers, increasing logarithmically from 29.38 m for the surface layer to 456.56 m for the deepest layer. The model underestimates the upwelling in the northeastern Atlantic, Arabian Sea and polar
regions (Ward et al., 2018) while it overestimates low-latitude upwellings (Ridgwell et al., 2007). However, on a global scale, Marsh et al. (2011) show that the model simulates realistic upwelling.

The atmospheric component (EMBM) is closely based on the UVic Earth system model (Weaver et al., 2001). It is a 2D model, where atmospheric temperature and specific humidity are the prognostic variables. Heat and moisture are horizontally transported by winds and mixing. The incoming shortwave radiation at the top of the atmosphere depends on the planetary
albedo, which varies as a function of latitude and time of the year to account for the effects of changes in solar zenith angle. The net longwave radiation represents $\sim 45\%$ of the total atmospheric energy balance while net shortwave radiation represents $\sim 25\%$. The radiative forcing associated with changes in atmospheric $CO_2$ concentrations is considered in the calculation of outgoing planetary longwave ($Q_{PLW}$). Higher atmospheric $CO_2$ concentration leads to higher amount of $Q_{PLW}$ being trapped in the atmosphere. Furthermore, the parameterization for $Q_{PLW}$ is taken from Thompson and Warren (1982) and depends on
the surface relative humidity and atmospheric temperature through a logarithmic dependency. Precipitation instantaneously removes all moisture corresponding to the excess above a relative humidity threshold. Wind velocities are prescribed following the annual average data of Trenberth (1989) and a constant and dimensionless land surface drag coefficient is set to $1 \times 10^{-3}$ (Weaver et al., 2001).



The sea-ice component (GOLDSTEINSEAICE) solves the fraction of the ocean surface covered by ice within a grid cell
and computes the average sea-ice thickness (Edwards and Marsh, 2005). A diagnostic equation is solved for the ice surface
temperature. Growth or decay of sea ice depends on the net heat flux into the ice (Hibler, 1979; Semtner, 1976). Sea-ice
dynamics consists of advection by surface currents and diffusion. The sea-ice component acts as a coupling module between
the ocean and the atmosphere, where heat and freshwater are exchanged and conserved between these three modules.

### 3.2 Ocean biogeochemistry component

The biogeochemical module (BIOGEM) represents the transformation and spatial redistribution of biogeochemical tracers
(Ridgwell et al., 2007). The state variables are inorganic nutrients and organic matter. Organic matter is partitioned into dis-
solved and particulate organic matter (DOM and POM). The model includes iron (Fe) and phosphate ($PO_4$) as limiting nutrients
but similar to Asselot et al. (2021), we do not consider nitrate ($NO_3^-$) here. Furthermore, BIOGEM calculates the air-sea $CO_2$
and $O_2$ exchange. These fluxes depend on the gas transfer velocity, the water density, the concentration of dissolved gas in the
ocean surface, the solubility coefficient calculated from Wanninkhof (1992), the concentration of gas in the atmosphere, and
the fraction of the ocean covered by sea ice (Ridgwell et al., 2007).

### 3.3 Ecosystem community component

The marine ecosystem component (ECOGEM) represents the marine plankton community and associated interactions within
the ecosystem (Ward et al., 2018). The biological uptake in ECOGEM is limited by light, temperature and nutrient availability.
Phytoplankton is allowed to flexibly take up nutrients according to availability. The production of dead organic matter is
a function of mortality and messy feeding. The surface production is then distributed along the water column as a depth-
dependent flux. To achieve this, the flux is partitioned between POM which is predominantly remineralized below 590 m deep,
and DOM which is remineralized above 590 m deep. This particular depth value has been calibrated against observations
following the ensemble Kalman filter method (Ridgwell et al., 2007). In ECOGEM, the sinking speeds of organic matter are
constant. The model assumes that photosynthesis is a Poisson function of irradiance and that phytoplankton growth is limited
though this function (Geider et al., 1998; Moore et al., 2001). The phytoplankton growth model requires $NO_3^-$ to simulate
chlorophyll synthesis but we do not consider this nutrient in our study. As a consequence, the nitrate biomass is equal to the
phosphate biomass times the standard Redfield ratio of 16 (Ward et al., 2018). Nutrient uptake is a Michaelis-Menten function
and phytoplankton growth is limited by a minimum function of internal nutrient status. Plankton biomass and organic matter are
subject to processes such as resource competition and grazing before being passed to DOM and POM. The ecosystem is divided
into different plankton functional types (PFTs) with specific traits. Each PFT can be sub-divided into size classes with specific
size-dependent traits. Yet we incorporate only two PFTs: one phytoplankton and one zooplankton species. Phytoplankton is
characterized by nutrient uptake and photosynthesis whereas zooplankton is characterized by predation traits. Zooplankton
grazing depends on the concentration of prey biomass and prey size, predominantly grazing on preys that are 10 times smaller
than themselves. The model considers nutrients (DIC, $PO_4$ and Fe), plankton biomass and organic matter (POM and DOM) as
state variables. However, plankton biomass is not subject to transport by oceanic circulation. ECOGEM considers a dynamic



photoacclimation (Geider et al., 1998) where chlorophyll-to-carbon ratio is regulated as the cell attempts to balance the rate of light capture by chlorophyll with the maximum potential rate of carbon fixation. Phytoplankton biomass can only be lost via grazing and mortality. Plankton mortality is reduced at very low biomass such that plankton cannot become extinct. The production of alkalinity is coupled to phytoplankton uptake of phosphate via a fixed linear ratio, meaning that alkalinity increases while phosphate is consumed. The exports of calcium carbonate ($CaCO_3$) and alkalinity are scaled to the export of POC via a spatially uniform value which is modified by a thermodynamically based relationship with the calcite saturation state. The dissolution of $CaCO_3$ below the surface is treated in a similar way to that of POM.

### 3.4 Temperature limitation

Metabolic processes of photosynthesis, nutrient uptake and zooplankton predation are all driven through the same exponential temperature limitation term (Ward et al., 2018). The temperature limitation scheme is given by Eq. 1:

$$\gamma_T = \exp(A \cdot (T - T_{ref}))$$ (1)

where $\gamma_T$ is the temperature limitation, $A$ is the temperature sensitivity ($0.05°C^{-1}$), $T$ is the sea surface temperature and $T_{ref}$ is the reference temperature. A reference temperature of 20°C is used because most experimentally determined metabolic rates are made at this temperature (Behrenfeld and Falkowski, 1997; Goldman, 1977; Rhee and Gotham, 1981). Photosynthesis is light limited, which results in a sub-exponential growth rate. Yet zooplankton predation disproportionately increases and nutrient uptake disproportionately decreases with increasing temperature, leading to limitation of photosynthesis and thus limitation of chlorophyll when temperatures exceed ∼20°C (Appendix A1).

### 3.5 Phytoplankton light absorption

In the original model version (Ward et al., 2018), light was only absorbed by phytoplankton. Following Asselot et al. (2021), a new light scheme is implemented where the absorbed light by phytoplankton is converted into heat and is able to affect the oceanic temperature. In the current model configuration, the incoming shortwave radiation varies seasonally. Moreover, the light level is calculated as the mean level of photosynthetically available radiation within each oceanic layer. The light penetrates until the sixth oceanic layer of the model (221.84 m), with maximum light absorption in the surface layer and minimum absorption in the sixth layer. The propagation of light within the ocean is limited by pure water and chlorophyll cells (Ward et al., 2018). The vertical light attenuation scheme is given by Eq. 2:

$$I(z) = I_0 \cdot \exp(-k_w \cdot z - k_{Chl} \cdot \int_0^z Chl(z) \cdot dz)$$ (2)

where $I(z)$ is the radiation at depth $z$, $I_0$ is the radiation at the surface of the ocean, $k_w$ is the light absorption by clear water ($0.04$ m$^{-1}$), $k_{Chl}$ is the light absorption by chlorophyll ($0.03$ m$^{-1}$(mg Chl)$^{-1}$) and $Chl(z)$ is the chlorophyll concentration



at depth $z$. The values for $k_w$ and $k_{Chl}$ are taken from Ward et al. (2018). The parameter $I_0$ is negative in the model because it is a downward flux from the sun to the surface of the ocean. Phytoplankton changes the optical properties of the ocean through phytoplankton light absorption, causing a radiative heating and changing the heat distribution in the water column (e.g. Wetzel et al., 2006; Anderson et al., 2007; Sonntag, 2013). We implement phytoplankton light absorption into the model (Eq. 3) following the scheme of Hense (2007) and Patara et al. (2012):

$$\frac{\partial T}{\partial t} = \frac{1}{\rho \cdot c_p} \frac{\partial I}{\partial z} \tag{3}$$

$\partial T/\partial t$ denotes the water temperature change only associated with radiative heating, $c_p$ is the specific heat capacity of water, $\rho$ is the ocean density, $I$ is the solar radiation incident at the ocean surface, and $z$ is depth. We assume that the whole light absorption heats the water (Lewis et al., 1983).

### 3.6   Model setup and simulations

We use the same model setup and parametrization as described in Asselot et al. (2021), with 32 oceanic vertical layers, primary production allowed until the sixth vertical layer (221.84 m deep) and incoming shortwave radiation varying seasonally. The ecosystem community is consistent with the community described in Asselot et al. (2021), with one phytoplankton species and one zooplankton species (Appendix B1). First, we run a 10,000 years spin-up with only BIOGEM to achieve a realistic distribution of nutrients. The spin-up is run with a constant pre-industrial atmospheric $CO_2$ concentration of 278 ppm. Second,

ECOGEM is switched on and the simulations are run for 736 years, representing the period between 1765 and 2500 (Meinshausen et al., 2011). In total we run 8 simulations with historical runs between 1765-2005 followed by RCP scenarios for the period between 2006 and 2500. The simulations are run with and without phytoplankton light absorption (Table 1). For the simulations without phytoplankton light absorption $k_{Chl} = 0$ m$^{-1}$(mg Chl)$^{-1}$ meaning that light is only attenuated by $k_w$ (Eq. 2). We run the simulations with prescribed global $CO_2$ emissions, which are the sum of the fossil, industrial and land-use

related $CO_2$ emissions (Figure 1). Moreover, all simulations include ECOGEM and are forced with the same constant flux of dissolved iron into the ocean surface (Mahowald et al., 2006). We compare the yearly-averaged outputs of the year 2500.





**Table 1.** Name and description of the simulations (PLA = phytoplankton light absorption).

| Name | Description |
| --- | --- |
| RCP2.6 | $CO_2$ emissions following RCP2.6 |
| RCP2.6-LA | $CO_2$ emissions following RCP2.6 with PLA |
| RCP4.5 | $CO_2$ emissions following RCP4.5 |
| RCP4.5-LA | $CO_2$ emissions following RCP4.5 with PLA |
| RCP6.0 | $CO_2$ emissions following RCP6.0 |
| RCP6.0-LA | $CO_2$ emissions following RCP6.0 with PLA |
| RCP8.5 | $CO_2$ emissions following RCP8.5 |
| RCP8.5-LA | $CO_2$ emissions following RCP8.5 with PLA |

## 3.7 Model inter-comparison

To validate our model setup, we compare our results with the results of an EMIC intercomparion (Zickfeld et al., 2013), which has a model setup close to our model setup. To be consistent with Zickfeld et al. (2013), we compute the surface atmospheric
temperature (SAT) increase between the periods 1986-2005 and 2281-2300, without phytoplankton light absorption. Independently of the RCP scenario, Figure 3 shows that our increases in SAT are in agreement with the global mean warming of Zickfeld et al. (2013) and lay in between the model ensemble minimum and maximum values. Thus, our model setup is suitable to study climate change.





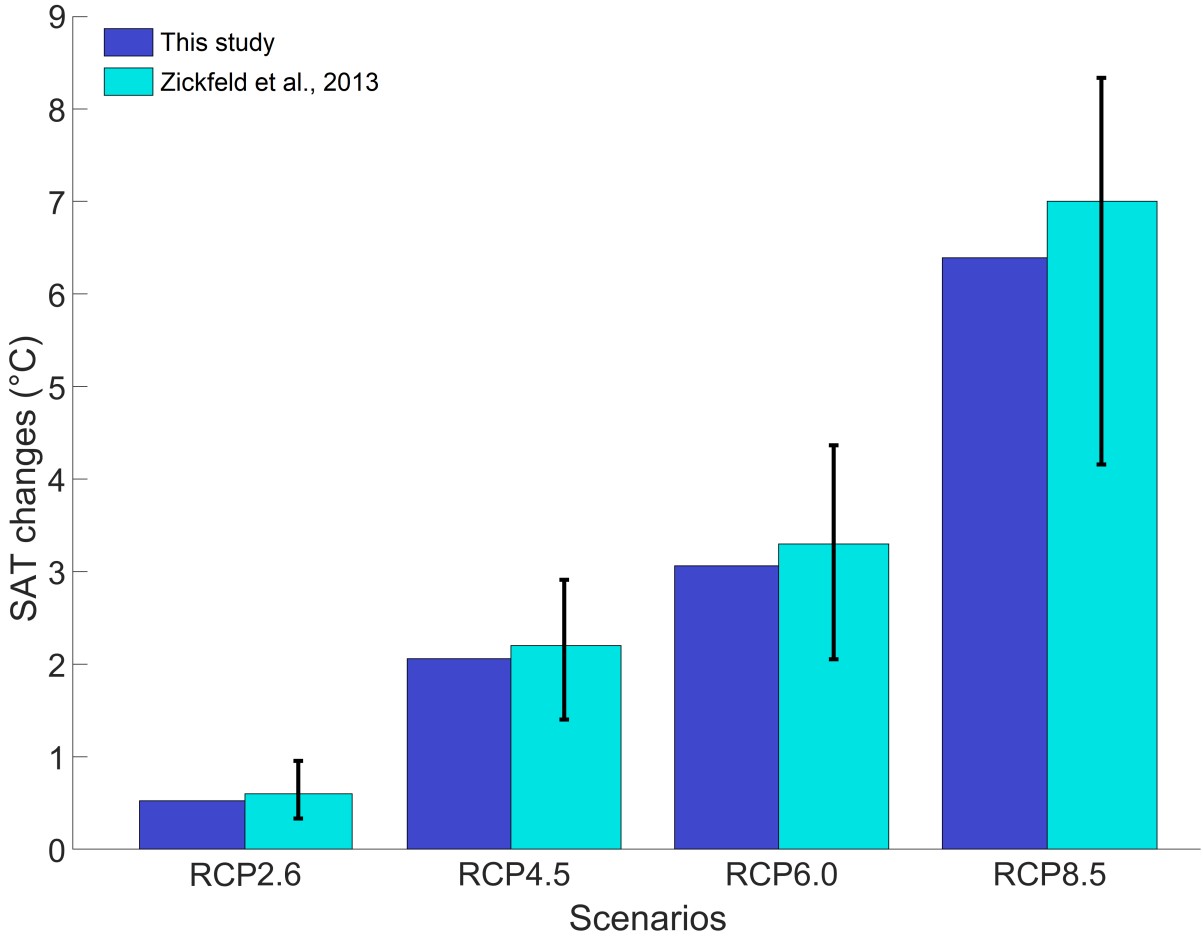

**Figure 3.** Global mean SAT changes (°C) between the periods 1986-2005 and 2281-2300 for our study (blue) and the study of Zickfeld et al. (2013) (turquoise). The black vertical lines represent the minimum and maximum values from the model inter-comparison of Zickfeld et al. (2013). The SAT changes of our study come from simulations without phytoplankton light absorption.

# 4   Results

In this section, we are interested in resolving the effects of phytoplankton light absorption and the relative differences between the simulations. Due to the limitations of such an EMIC, the absolute values are less relevant. We first look at the ocean properties such as the biological pump, surface chlorophyll and SST. Second, we investigate the changes in atmospheric $CO_2$ concentrations and SAT.





## 4.1 Oceanic properties

### 4.1.1 The biological carbon pump

To compare the strength of the biological carbon pump between our simulations, we look at vertical fluxes of POC in the water column. In our study, these fluxes are originally described by an exponential decay. However, to compare the vertical POC fluxes we compute them via a Martin curve (Martin et al., 1987), which is often used as a diagnostic tool for comparison. Even if our modeled POC fluxes follow an exponential decay function, using a Martin's curve function to compare these fluxes is

a reasonable assumption because these two functions give similar vertical POC profiles in our model (Ridgwell, 2001). The Martin curve is a power-law function where the dimensionless exponent (b) indicates the strength of the biological carbon pump. Large b-values indicate that organic matter is remineralised predominantly at shallow depths, highlighting a weak biological carbon pump, while low b-values indicate that remineralisation happens deeper in the water column, highlighting a strong biological carbon pump. Independently of the RCP scenario, phytoplankton light absorption increases the b-values for

remineralisation rate, indicating that organic matter is remineralised at shallower depth with this biogeophysical mechanism (Table 2). The enhanced surface remineralization with phytoplankton light absorption is due to the higher amount of organic matter, generated by the higher primary production at the ocean surface (Table 2). The biological pump is therefore weaker with phytoplankton light absorption meaning that more inorganic matter, such as nutrients are located in the surface of the ocean.

**Table 2.** Comparison of b-values (no units) for remineralisation rate and primary production (Gt/yr) in the first oceanic layer. Note that phytoplankton light absorption always increases b-values and primary production.

| Simulation | b-values | Primary production (Gt/yr) |
|---|---|---|
| RCP2.6 | 0.6108 | 37.42 |
| RCP2.6-LA | 0.6212 | 38.15 |
| RCP4.5 | 0.6102 | 37.78 |
| RCP4.5-LA | 0.6203 | 38.31 |
| RCP6.0 | 0.6105 | 37.80 |
| RCP6.0-LA | 0.6195 | 38.49 |
| RCP8.5 | 0.6112 | 38.35 |
| RCP8.5-LA | 0.6177 | 38.67 |

### 4.1.2 Surface chlorophyll

We look at the distribution of chlorophyll at the ocean surface because this climate variable directly affects the heat distribution along the water column through phytoplankton light absorption. On a global scale, independently of the RCP scenario, phytoplankton light absorption leads to an increase of chlorophyll at the ocean surface (Figure 4). This increase is due to two mechanisms. First, phytoplankton light absorption leads to a weaker biological pump (Table 2). As a consequence, more labile



organic matter lays in the ocean surface, increasing the remineralization and thus the surface nutrient concentrations. Second, phytoplankton light absorption leads to a differential heating between the surface and bottom of the ocean. The ocean surface experiences a stronger heating than the ocean bottom due to the direct effect of phytoplankton light absorption at the surface while heat is slowly transported and redistributed at the bottom of the ocean by oceanic circulation. As a consequence, the pressure gradient along the water column is strengthened and the upward vertical velocity is enhanced (Appendix D1), bringing

more nutrients at the ocean surface. The increased surface nutrient concentrations (Appendix E1) via these two mechanisms lead to the higher surface chlorophyll with phytoplankton light absorption. For the RCP2.6, RCP4.5 and RCP6.0 scenarios, the global increase of chlorophyll is between 0.015 and 0.019 mgChl/m$^3$, representing an increase of 13-15%. These assessments are slightly higher than previous estimates showing an increase between 4 and 12% (Manizza et al., 2005; Asselot et al., 2021). However, compared to our model setup, Manizza et al. (2005) use an ocean model, neglecting any interactions between

the ocean and the atmosphere. Additionally, Asselot et al. (2021) do not prescribe CO$_2$ emissions, neglecting the changes in chlorophyll due to climate change. The increase of chlorophyll for the RCP8.5 scenario is the smallest, with an increase of ~0.01 mgChl/m$^3$, representing an increase of 8%. The lower global increase of chlorophyll under RCP8.5 compared to the other RCPs scenarios is due to the lower increase of chlorophyll in the mid-latitudes and upwelling regions (Figure 5).

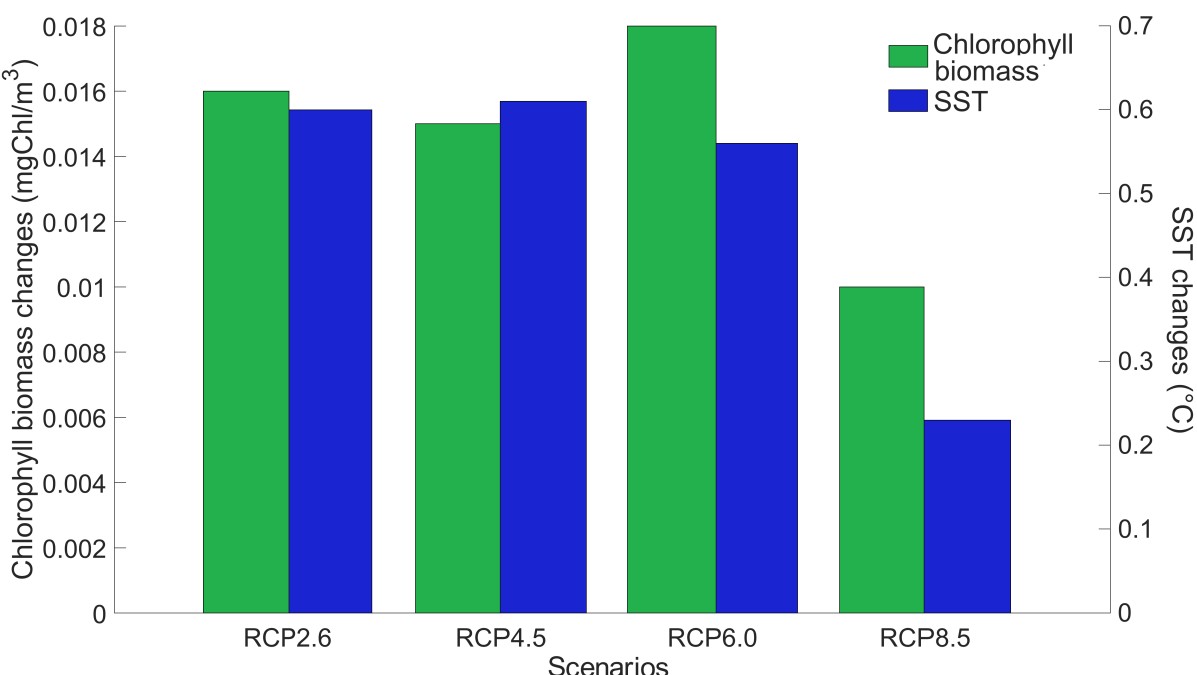

**Figure 4.** Globally-averaged surface chlorophyll (mgChl/m$^3$) and SST (°C) changes between the RCP scenarios. The values represent the difference between the simulation with minus without phytoplankton light absorption. Note that the y-axis scales are always positive, indicating that phytoplankton light absorption always leads to a global increase of surface chlorophyll and SST.



The regional patterns of surface chlorophyll changes due to phytoplankton light absorption are similar between the RCP scenarios (Figure 5). The largest differences of chlorophyll occur in the high latitudes. Such as, between the simulations *RCP8.5-LA* and *RCP8.5*, the maximum increase of 0.4 mgChl/m$^3$ occurs in the northern polar region (Figure 5d). This pronounced chlorophyll response in the high latitudes is explained by two mechanisms: First, chorophyll is not subject to transport and therefore cannot be redistributed in the mid-latitudes. Second, light availability is enhanced for phytoplankton growth due to the decrease of sea-ice. For instance, the global sea-ice area decreases by 13% between *RCP8.5-LA* and *RCP8.5*, thus increasing light availability for phytoplankton growth. The upwelling and mid-latitude regions show a higher chlorophyll concentration with phytoplankton light absorption. These regional patterns are due to enhanced vertical velocity caused by the differential heating between the surface and bottom of the ocean, strengthening the vertical pressure gradient. For instance, in the upwelling region along the Chilean coast, at 115 m depth, the global vertical velocity is enhanced by 10.3% in *RCP4.5-LA* compared to *RCP4.5* (Appendix D1). As a result, on a global scale, more nutrients are brought to the surface, decreasing the nutrient limitation and thus promoting a higher phytoplankton biomass at the surface. In contrast to the upwelling regions, the subtropical gyres show no or small differences in surface chlorophyll concentrations.





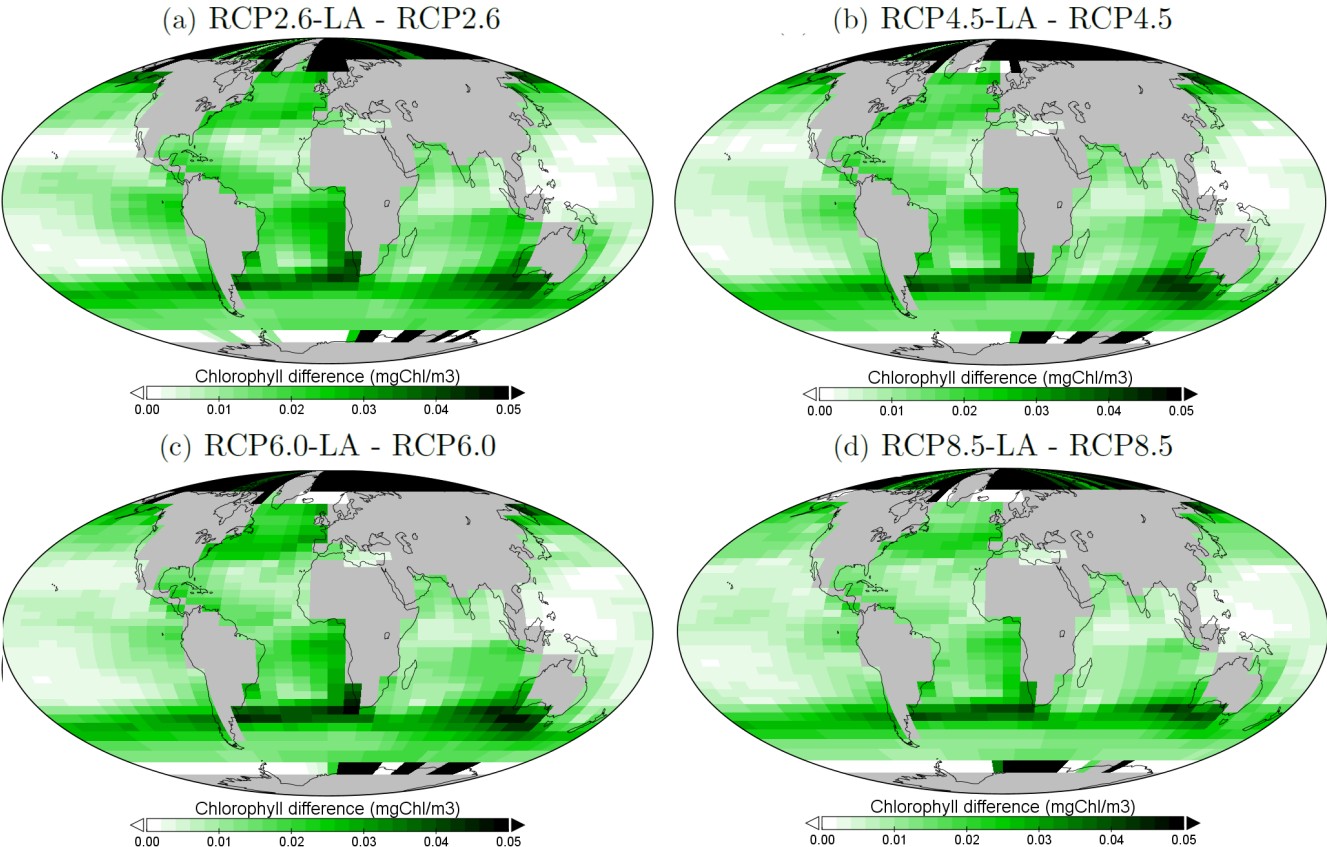

**Figure 5.** Chlorophyll changes at the surface (mgChl/m$^3$) for the different simulations. (a) Difference between RCP2.6-LA and RCP2.6. (b) Difference between RCP4.5-LA and RCP4.5. (c) Difference between RCP6.0-LA and RCP6.0. (d) Difference between RCP8.5-LA and RCP8.5. The scale and color coding are identical between the four panels. Note that the scale is logarithmic and always positive.

### 4.1.3 Sea surface temperature

Due to changes in surface chlorophyll, we expect variations in SST. Our results highlight that under the RCP2.6, RCP4.5 and RCP6.0 scenarios, phytoplankton light absorption increases the SST by ∼0.6°C (Figure 4). These assessments are higher than previous global estimates, giving a global SST increase of 0.33-0.5°C (Wetzel et al., 2006; Patara et al., 2012; Asselot et al., 2021). This stronger increase in SST is caused by higher increases in surface chlorophyll compared to previous assessments. For the RCP8.5 scenario, phytoplankton light absorption only increases SST by 0.23°C. This lower increase in SST is due to the lower increase in global surface chlorophyll under this scenario. The regional patterns of SST changes due to phytoplankton light absorption are similar between the simulations but the magnitude of changes differs (Figure 6). Independently of the RCP scenario, even if the polar regions experience a high increase in chlorophyll, they also experience the lowest increase of SST. This is due to the underestimated oceanic circulation in these regions, which is due to the coarse grid resolution, limiting the

 

heat redistribution. For instance, between the simulations *RCP4.5-LA* and *RCP4.5*, the minimum increase of 0.03°C occurs in the Southern Ocean. Even in the regions where small differences in surface chlorophyll occur, such as the subtropical gyres, we find high SST increases. The missing spatial patterns between chlorophyll and SST can be explained by the model setup.

Chlorophyll is not subject to transport while physical quantities, such as heat, are transported by oceanic currents. Therefore, heat is smoothly redistributed around the globe.

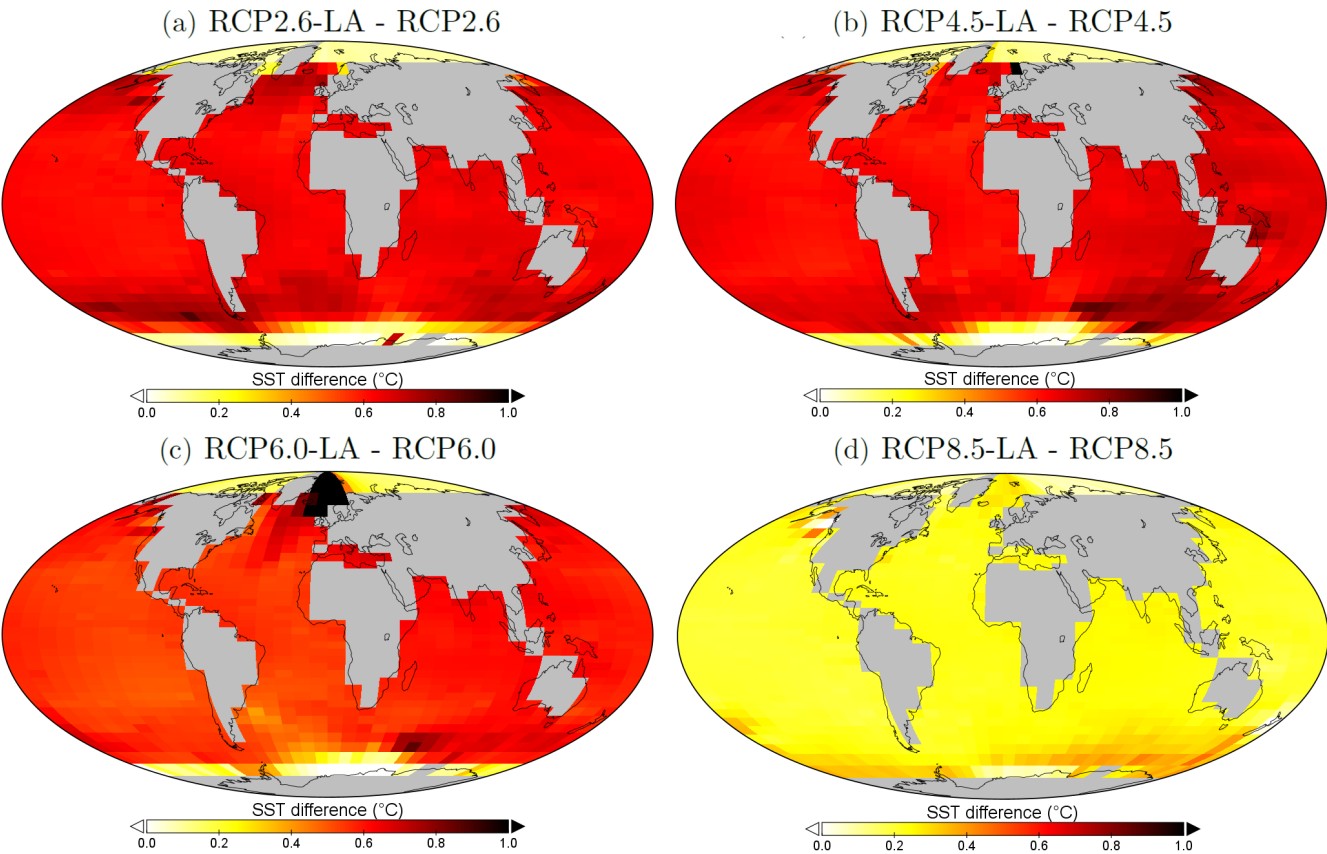

**Figure 6.** Sea surface temperature changes (°C) between the simulations. (a) Difference between RCP2.6-LA and RCP2.6. (b) Difference between RCP4.5-LA and RCP4.5. (c) Difference between RCP6.0-LA and RCP6.0. (d) Difference between RCP8.5-LA and RCP8.5. The scale and color coding are identical between the four panels. Note that the scale is always positive.

## 4.2 Atmospheric properties

### 4.2.1 Atmospheric CO$_2$ concentration

Even though we prescribe RCP emissions in our simulations, the atmospheric CO$_2$ concentrations in our results do not match
the projected atmospheric concentrations in 2500 of Meinshausen et al. (2011). Our version of the model, has been tuned to get



reasonable primary production and nutrient fields but not to get future atmospheric $CO_2$ concentrations. As a consequence, with this configuration, the model is known to simulate low atmospheric $CO_2$ concentrations (Asselot et al., 2021, 2022). Because we are more interested in qualitative assessment rather than quantitative estimates, such limitation does not affect the main findings of our study. Independently of the RCP scenario, the atmospheric $CO_2$ concentration increases with phytoplankton
light absorption (Figure 7). For the RCP2.6, RCP4.5 and RCP6.0 scenarios, phytoplankton light absorption increases the atmospheric $CO_2$ concentration by ∼20% while a previous study indicates an increase of 10% (Asselot et al., 2021). However, Asselot et al. (2021) do not prescribe $CO_2$ emissions, neglecting their effect on the atmospheric $CO_2$ concentration. For the RCP8.5 scenario, the atmospheric $CO_2$ concentration increases by 8% only, which is due to the lower increase in chlorophyll and SST.

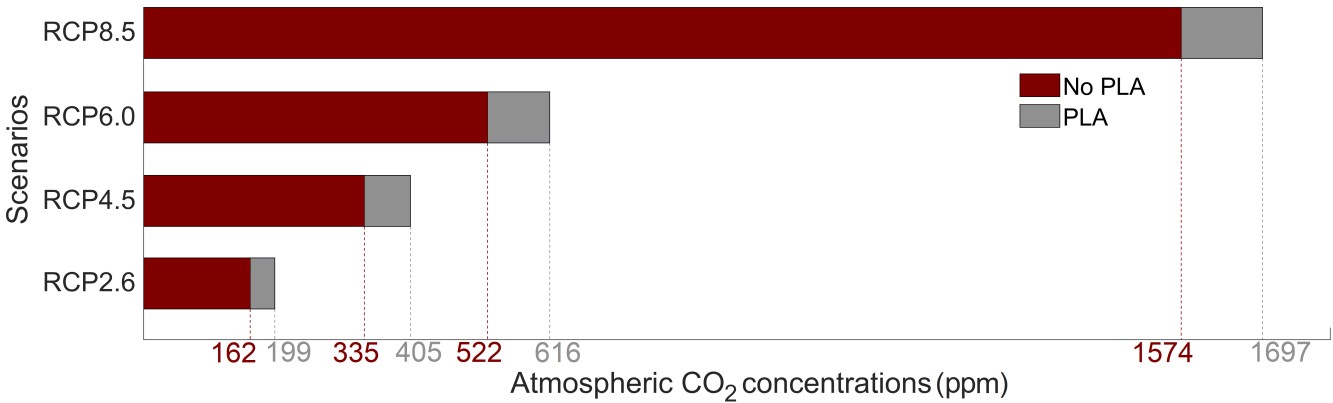

**Figure 7.** Atmospheric $CO_2$ concentrations (ppm) for the 8 simulations at year 2500. PLA stands for phytoplankton light absorption.

The increase in atmospheric $CO_2$ concentrations with phytoplankton light absorption is mainly due to the higher SST decreasing $CO_2$ solubility, and thus increasing the oceanic $CO_2$ outgassing (Asselot et al., 2022). The reduced solubility pump enhances the ocean-to-atmosphere $CO_2$ flux by ∼10%. In contrast, the changes in the biological and carbonate pump enhance the air-sea $CO_2$ fluxes by <1%. However, the temperature dependence of solubility could not explain the changes in atmospheric $CO_2$ concentration in steady state, suggesting that the effect is transient. For the first three RCPs scenarios (but not
RCP8.5), the change in atmospheric $CO_2$ concentration driven by phytoplankton light absorption follows a roughly linear dependence on the baseline concentration for that RCP (Figure 8). The rate of $CO_2$ uptake is roughly proportional to baseline concentration for the first three RCPs scenarios but is reduced for RCP8.5 because of the smaller effect of phytoplankton light absorption on SST. To validate this inference, we continue our simulations for another 1000 years with no further $CO_2$ emissions (Appendix C1). These additional simulations indicate that $CO_2$ differences decrease through time, converging towards
the far smaller steady-state difference previously highlighted by Asselot et al. (2021).





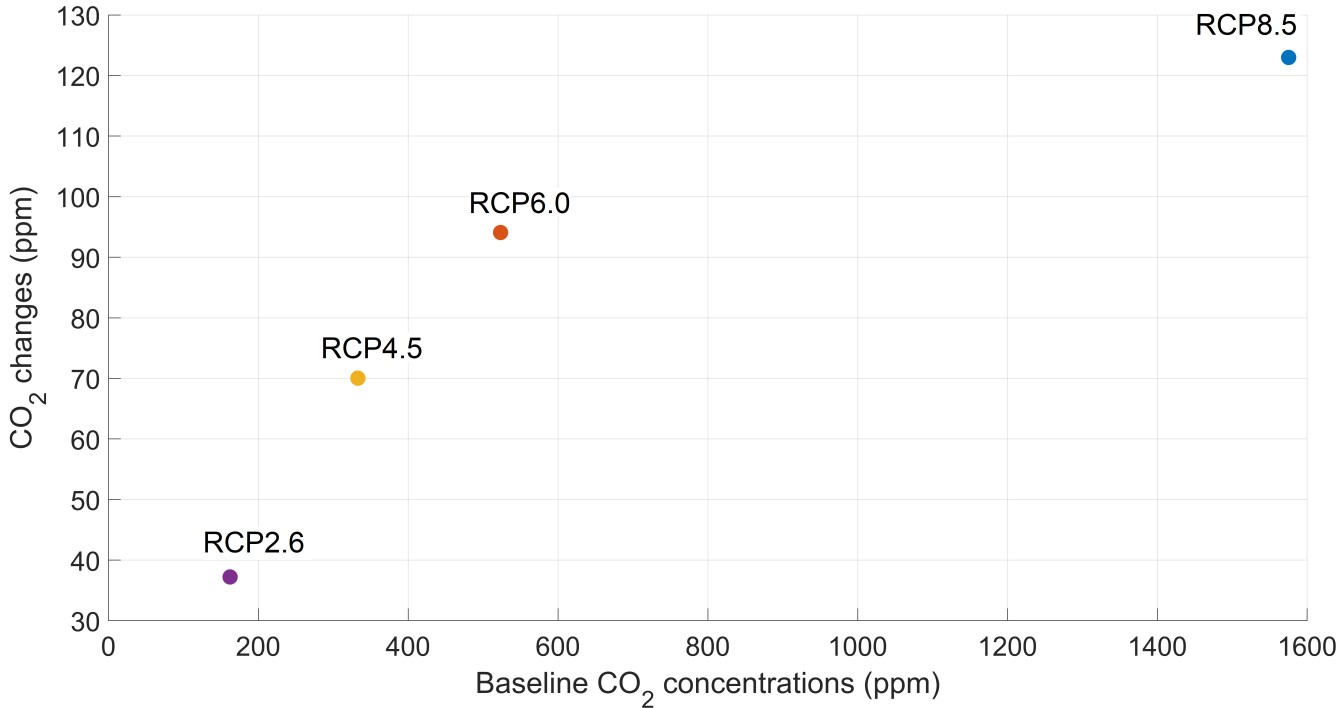

**Figure 8.** Phytoplankton light absorption driven $CO_2$ changes (ppm) under the four RCPs scenarios. The x-axis represents the atmospheric $CO_2$ concentrations of the simulations without phytoplankton light absorption.

### 4.2.2 Surface atmospheric temperature

Due to higher greenhouse gases concentrations, the atmospheric temperature increases with phytoplankton light absorption (Figure 9). For the RCP2.6, RCP4.5 and RCP6.0 scenarios, the global increase in SAT is $\sim$0.8°C, which is higher than previous model estimates indicating a zonally-averaged SAT increase of 0.2-0.45°C (Shell et al., 2003; Patara et al., 2012; Asselot et al.,

2021). However, compared to our model setup, Shell et al. (2003) use an uncoupled ocean-atmosphere model, neglecting any interactions between the ocean and the atmosphere. Patara et al. (2012) use a constant and prescribed atmospheric $CO_2$ concentration for their simulations, neglecting its effect on the atmospheric temperature. Asselot et al. (2021) do not prescribe $CO_2$ emissions, neglecting changes in the heat budget due to climate change. With a value of 0.28°C, the increase in SAT under the RCP8.5 scenario is lower than for the other RCP scenarios. This lower value is driven by a combination of reduced SST

warming and lower atmospheric $CO_2$ concentration changes under this RCP scenario.





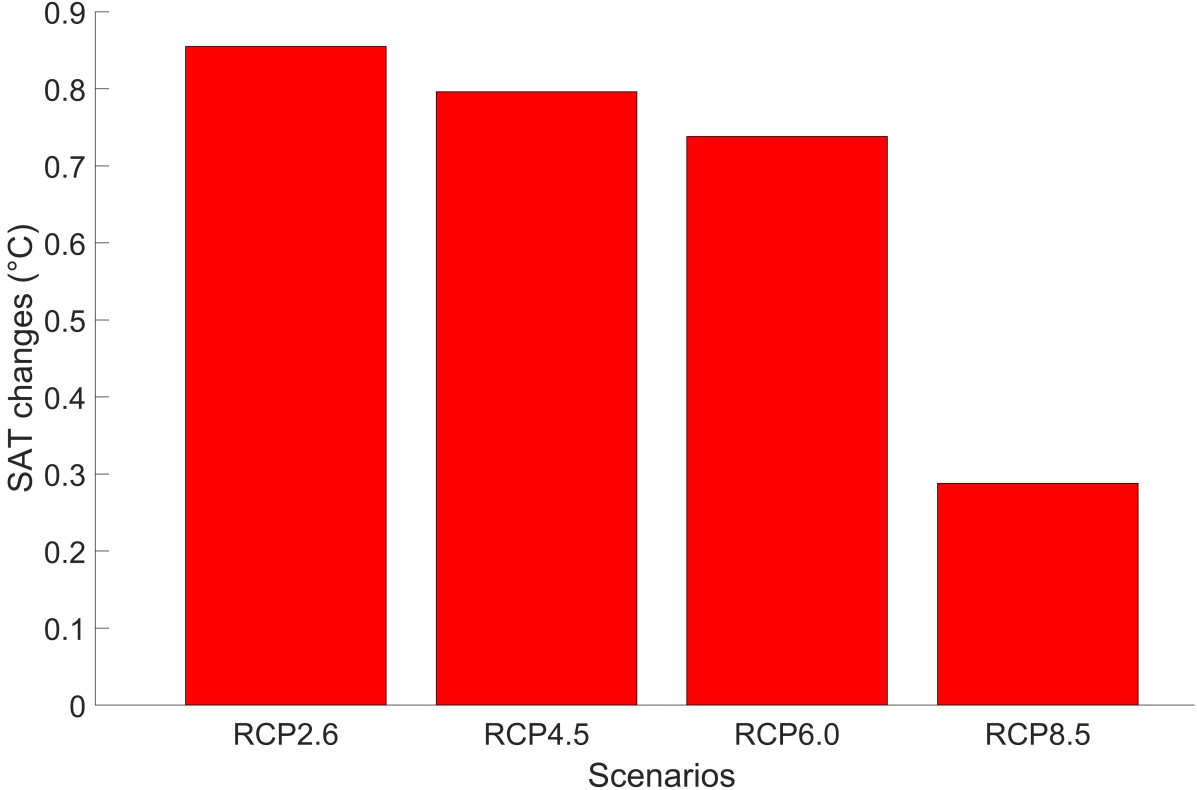

**Figure 9.** Globally-averaged SAT (°C) changes between the RCP scenarios. The values represent the difference between the simulation with and without phytoplankton light absorption. Note that the y-axis scale is always positive, indicating that phytoplankton light absorption always leads to a global increase of SAT.

The regional patterns of SAT changes due to phytoplankton light absorption are similar among the RCP scenarios but the magnitude of changes differs (Figure 10). The polar regions experience a strong increase in SAT, with the highest values occurring in the Southern Ocean. For instance, comparing the simulations *RCP4.5-LA* and *RCP4.5*, the maximum increase of 1.6°C occurs in the Southern Ocean (Figure 10b). This maximum value is due to the rather coarse grid resolution in the high latitudes. This estimate is again higher than previous local estimates (Shell et al., 2003; Patara et al., 2012; Asselot et al., 2021) for the same reasons described above. Furthermore, around the rest of the globe, heat is redistributed smoothly in the atmosphere.




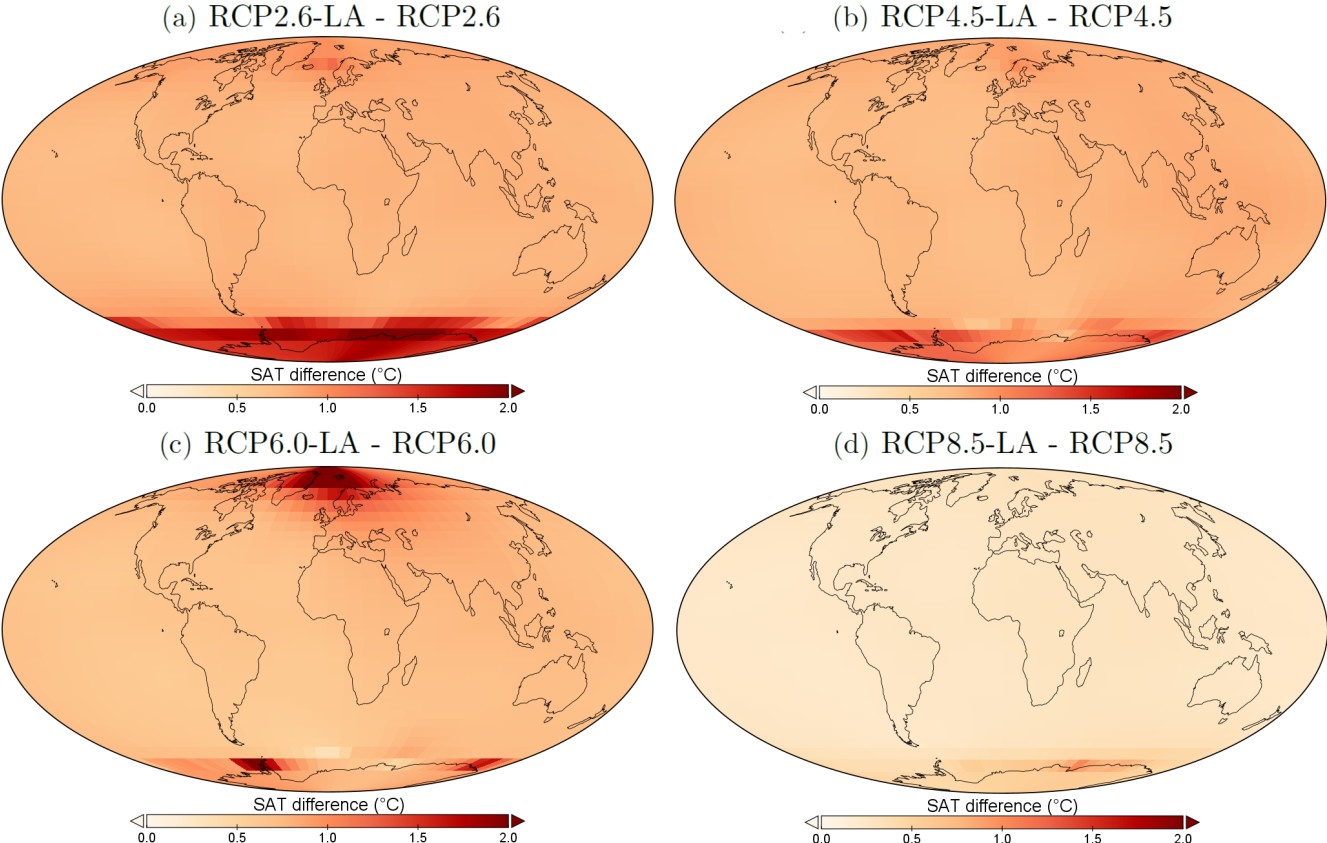

**Figure 10.** Surface atmospheric temperature changes (°C) between the different simulations. (a) Difference between RCP2.6-LA and RCP2.6. (b) Difference between RCP4.5-LA and RCP4.5. (c) Difference between RCP6.0-LA and RCP6.0. (d) Difference between RCP8.5-LA and RCP8.5. The scale and color coding are identical for the four panels. Note that the scale is always positive.

## 5   Discussion and conclusions

### 5.1   General discussion

Our results show that phytoplankton light absorption affects water temperature and nutrient concentrations. The increase in surface nutrient concentrations (Appendix E1) is due to two mechanisms. First, phytoplankton light absorption leads to a weaker biological pump, increasing the remineralization at the ocean surface. Second, phytoplankton light absorption leads to a differential heating between the surface and bottom of the ocean, strengthening the vertical pressure gradient and thus enhancing the upward vertical circulation, especially in the upwelling regions. The increased surface nutrient concentrations leads to higher

surface chlorophyll, which in turn leads to a warming of the ocean surface. Furthermore, the higher $CO_2$ concentration with phytoplankton light absorption leads to an enhanced greenhouse gas effect. As a consequence, the radiative forcing increases,





warming the ocean surface as well. Under the RCP2.6, RCP4.5 and RCP6.0 scenarios, phytoplankton concentration is not strongly limited by temperature. As a result the impact of phytoplankton light absorption on the climate system is similar between these RCP scenarios. However, under the RCP8.5 scenario, the effect of phytoplankton light absorption on the climate

system is reduced. This is due to the model setup where a SST higher than ∼20°C limits net phytoplankton concentration (Appendix A1). This limit is exceeded in all the simulations but *RCP8.5-LA* and *RCP8.5* are the only ones where the average SST exceeds 20°C (Appendix F1). Phytoplankton concentration is thus limited by temperature and the difference of chlorophyll between *RCP8.5-LA* and *RCP8.5* is weaker than between the other simulations (Figure 4). The response of the climate system to phytoplankton light absorption is therefore weaker under the RCP8.5 scenario. Our findings indicate that the effect of

phytoplankton light absorption is smaller under high greenhouse gas emissions compared to reduced and intermediate greenhouse gas emissions. In agreement with Patara et al. (2012), this study indicates that a severely warmer world increases ocean clarity and slows down the warming due to phytoplankton light absorption. However, the reduced effect of phytoplankton light absorption under the RCP8.5 scenario may not be as strong if phytoplankton were able to adapt to higher temperature in our model setup.

## 5.2  Limitations

For the first time, using EcoGEnIE (Ward et al., 2018), we investigate the impact of phytoplankton light absorption under prescribed $CO_2$ emissions following the RCP scenarios on a multi-century timescale. However our model setup has limitations that must be overcome to improve our quantitative estimates. Most notably, our version of the model must be tuned to fit the projected atmospheric $CO_2$ concentrations under global warming scenarios. For instance, for the simulations following

the RCP2.6 scenario, the final atmospheric $CO_2$ concentrations and SSTs are lower than pre-industrial levels. This is due to the negative emissions for this scenario and the underestimation of the atmospheric $CO_2$ concentrations with our model setup (Asselot et al., 2021, 2022). As detailed previously, primary production is allowed until the sixth oceanic layer and the model has not been tuned in this configuration yet. The lower levels under the RCP2.6 scenario compared to the pre-industrial levels are not an issue for our study because we exclusively focused on the effect of phytoplankton light absorption

rather than on the differences between the simulations and the pre-industrial state. Furthermore, we switch on ECOGEM and the RCP emission forcings at the same time. We know from previous work (Asselot et al., 2021), that switching on ECOGEM decreases the atmospheric $CO_2$ concentration, thus our simulations contain an effect of both drift and emissions. However, the drifting effect is identical between simulations and therefore balances out when comparing simulations. The model inter-comparison against Zickfeld et al. (2013) suggests that this drifting effect is not an issue because the response

to ECOGEM is fast enough that most of the adjustment happen in the first 200 years of simulation. With our model setup we demonstrate that phytoplankton light absorption increases local SST by 0.4-1.1°C depending on the scenario considered. These estimates are lower than previous observations showing a local increase of SST by 0.95-4.5°C (Kahru et al., 1993; Capone et al., 1998; Wurl et al., 2018). The difference between our estimates and observations may in part be due to the short time scales for observations while our estimates are yearly-averaged. Our results indicate a large local increase in chlorophyll in

the simulations with phytoplankton light absorption, especially in the northern polar region. However, plankton biomass is not



subject to transport by ocean currents. If phytoplankton biomass would be advected, we suppose that these local increase would be smaller. Additionally, if wind stress could evolve freely, we suppose that the increase in atmospheric temperature would lead to an increased wind stress. As a result, the upwelling dynamics would be enhanced. In our model setup, temperature significantly affects the concentration of our bulk phytoplankton. The temperature dependent grazing that leads to increased
grazing pressure as well as the temperature dependent nutrient uptake that leads to increased nutrient limitation with increasing temperature results in a decrease in phytoplankton concentration when oceanic temperature exceeds about 20°C. The response of the modeled phytoplankton might be different if we would have consider different PFTs (e.g. diatoms, dinoflagellates) since their concentrations are characterized by different temperature response curves (Anderson et al., 2021). We cannot rule out that the strong phytoplankton concentration limitations in our simulations *RCP8.5-LA* and *RCP8.5* will also occur if more PFTs
were considered. Depending on the model and on the region of interest, the future of primary production is highly uncertain. For instance, using a suite of nine coupled carbon-climate ESMs under the RCP8.5 scenario, Laufkötter et al. (2015) show that primary production may increase, remain stable or decrease under global warming. Though we note that the simulations of Laufkötter et al. (2015) only went out to 2100, not to 2500 as in our extended simulations. Our results highlight that phytoplankton light absorption itself increases chlorophyll leading to more heat being trapped in the ocean surface.

## 5.3   Implication for Earth system models

The traditional view is that dominant carbon cycle uncertainties come from the terrestrial response to elevated atmospheric $CO_2$ concentrations. For instance, the net land emissions over the 1858-2008 period is estimated as likely (66% confidence) to lie in the range from 0 to 128 GtC (Holden et al., 2013). However, this work suggests that introducing biogeophysical mechanisms such as phytoplankton light absorption leads to major carbon cycle uncertainties. For instance, with our model
setup, implementing phytoplankton light absorption increases the atmospheric carbon content by 79 GtC in RCP2.6 and by 258 GtC in RCP8.5. This study highlights a highly uncertain feedback on the carbon cycle that is missing from 50% of the CMIP6 models (Pellerin et al., 2020). Neglecting the effect of phytoplankton light absorption on the carbon cycle can lead to incomplete future climate projections, thus this biogeophysical mechanisms should be included by default in climate models.

*Code availability.*   The code for the model is hosted on GitHub and can be obtained by cloning or downloading: https://zenodo.org/record/5676165.
The configuration file is named "RA.ECO.ra32lv.FeTDTL.36x36x32" and can be found in the directory "EcoGENIE_LA/genie-main/configs". The user-configuration files to run the experiments can be found in the directory "EcoGENIE_LA/genie-userconfigs/RA/Asselotetal_ESD". Details of the code installation and basic model configuration can be found on a PDF file (https://www.seao2.info/cgenie/docs/muffin.pdf). Finally, section 9 of the manual provides tutorials on the ECOGEM ecosystem model.





*Author contributions.* All authors designed and developed the concept of the study. RA performed the analysis of the model outputs with inputs from IH. RA drafted the initial version of the manuscript in collaboration with IH. All co-authors read and reviewed the final version of the manuscript.

*Competing interests.* The authors declare that they have no conflict of interest.

*Acknowledgements.* Our special thanks go to Jana Hinners, Isabell Hochfeld, Félix Pellerin, Maike Scheffold and Laurin Steidle for their valuable comments on the early version of this manuscript. This work was supported by the Center for Earth System Research and Sustainability (CEN), University of Hamburg, and contributes to the Cluster of Excellence "CLICCS - Climate, Climatic Change, and Society".





## Appendix A:  Optimum net phytoplankton growth

To illustrate the temperature limitation, we show the effect of SST on surface chlorophyll. Appendix A1 shows that a decrease in surface chlorophyll happens around 20°C, indicating the limiting factors of nutrient demand and zooplankton predation begin to dominate at around 20°C.

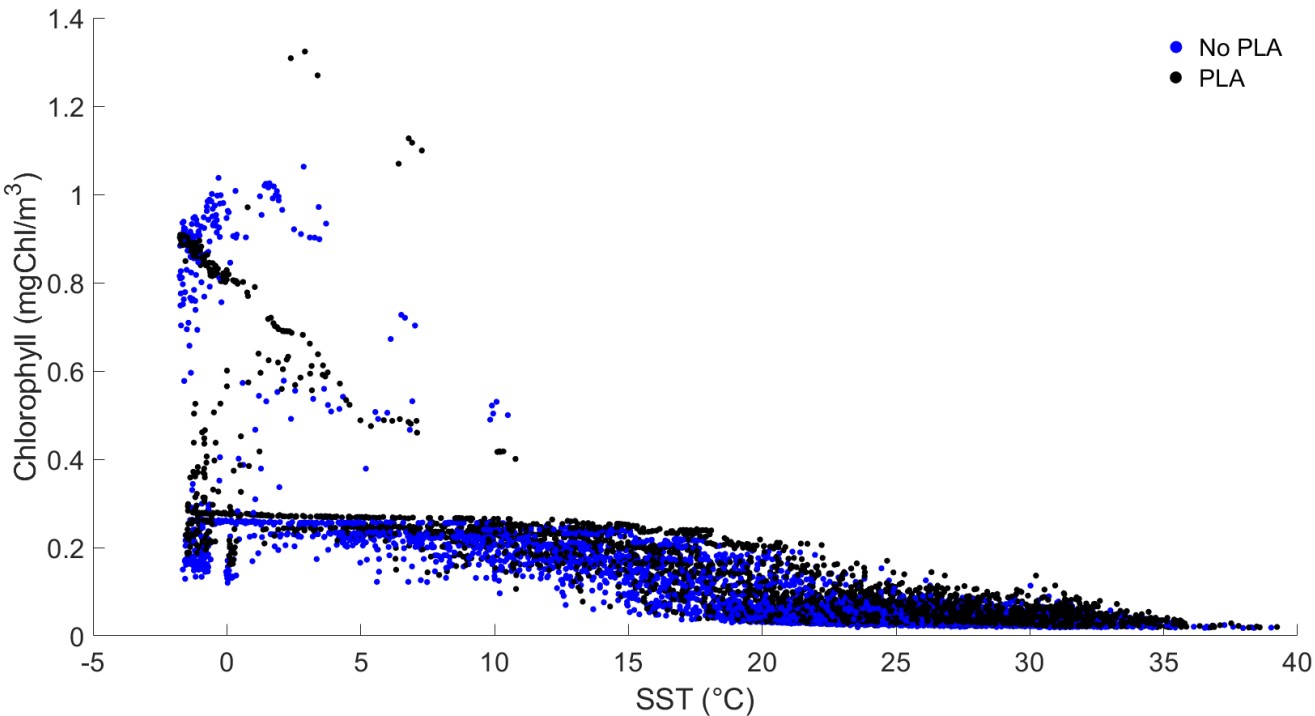

**Figure A1.** Effect of SST (°C) on surface chlorophyll (mgChl/m$^3$) for each grid cell. Blue color represents the four simulations without phytoplankton light absorption (PLA) while black color represents the four simulations with PLA.



## Appendix B: Plankton functional types


We base our ecosystem community on the one described by Ward et al. (2018). We only use 2 PFTs: one phytoplankton group and one zooplankton group (Appendix B1). We show that the complexity of the ecosystem does not have an important impact on the climate system compared to the effect of phytoplankton light absorption (Asselot et al., 2021). Therefore, for simplification, we reduce the ecosystem complexity.

**Table B1.** Size of the different plankton functional types ($\mu$m) used during the simulations.

| PFT | Size ($\mu$m) |
|---|---|
| Phytoplankton | 46.25 |
| Zooplankton | 146.15 |




## Appendix C:  Additional simulations

To investigate the substantially reduced ocean $CO_2$ uptake with phytoplankton light absorption, we continue our simulations for another 1000 years with no further $CO_2$ emissions. The difference in atmospheric CO2 concentrations between the simulations with and without phytoplankton light absorption decreases through time. This result evidences that large $CO_2$ differences are driven by a transient effect of reduced $CO_2$ uptake fluxes, consistent with reduced $CO_2$ solubility under phytoplankton light absorption warming.

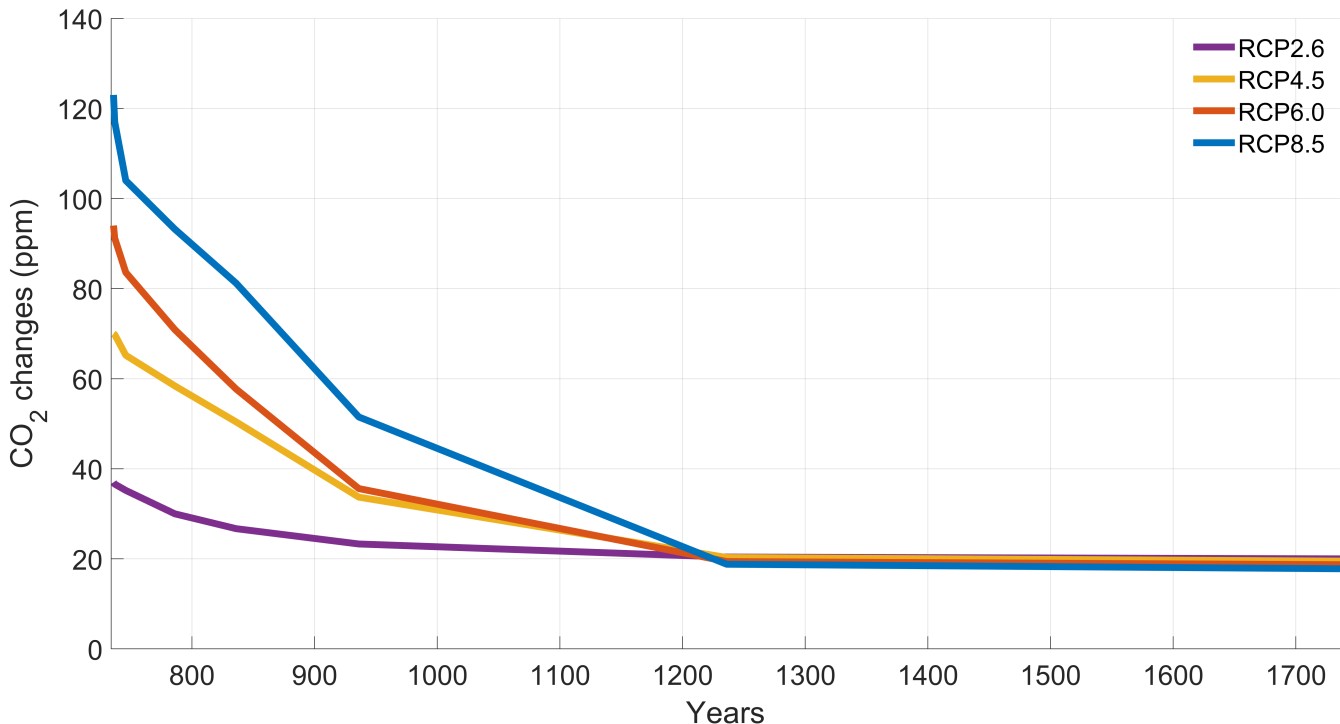

**Figure C1.** Difference in atmospheric $CO_2$ concentrations through time for the additional simulations.





## Appendix D: Upward vertical velocity

To illustrate the changes in upward vertical velocity with phytoplankton light absorption, we show the velocities for all the simulations in the Chilean upwelling region. Independently of the RCP scenario, phytoplankton light absorption leads to an enhanced vertical velocity in the region. As a result, a larger amount of nutrients are brought to the ocean surface, leading in part to the higher chlorophyll with phytoplankton light absorption.


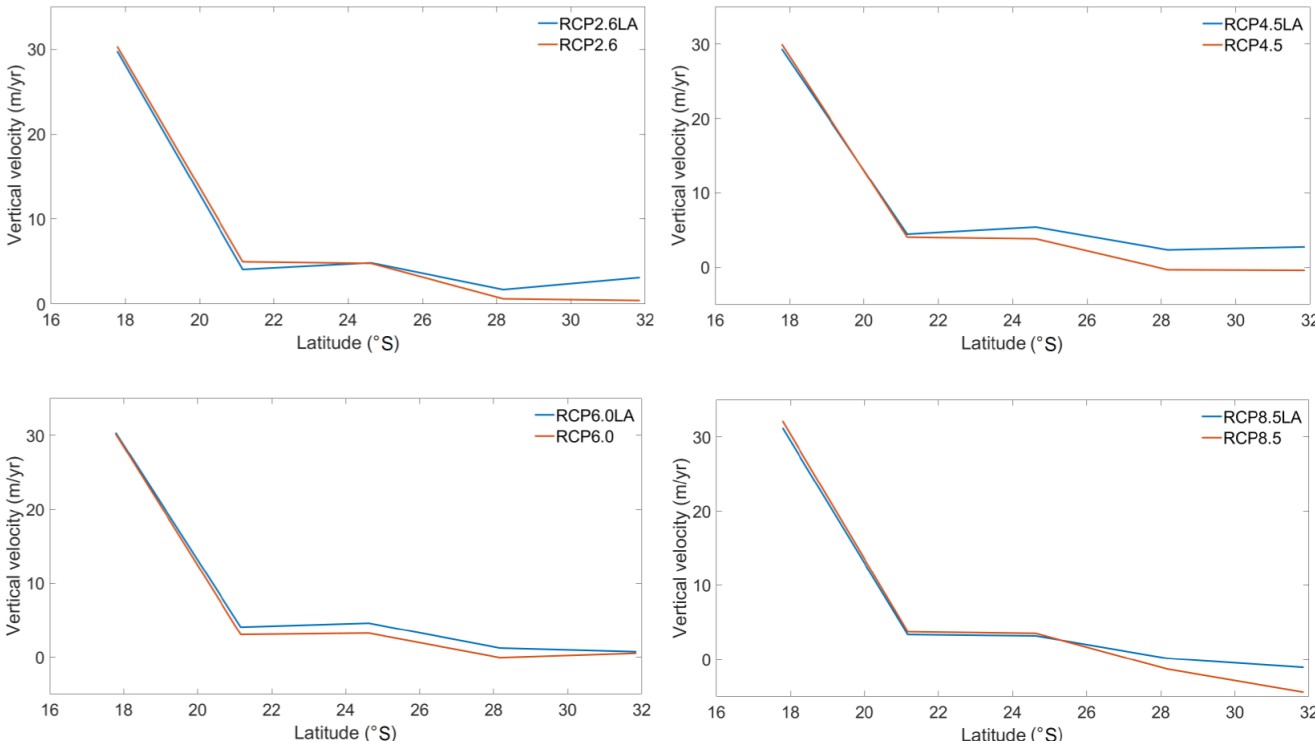

**Figure D1.** Vertical upward velocity (m/yr) at 115 m depth in the upwelling-region along the South American coast. Blue lines represent the simulations with phytoplankton light absorption while the red lines represent the simulations without.



**Appendix E:  Surface phosphate concentration**

Independent of the RCP scenarios, our results evidence an increase in surface nutrients, such as phosphate. As a result, the surface chlorophyll biomass increases with phytoplankton light absorption.

**Table E1.** Phosphate concentration changes at the surface (mol/kg) for the different RCP scenarios. The values represent the difference with minus without phytoplankton light absorption. The "+" symbol indicates an increase in surface $PO_4$ concentration.

| Scenario | $\Delta PO_4$ conc. (mol/kg) |
|---|---|
| RCP2.6 | $+8.9 \cdot 10^{-8}$ |
| RCP4.5 | $+8.6 \cdot 10^{-8}$ |
| RCP6.0 | $+9.1 \cdot 10^{-8}$ |
| RCP8.5 | $+7.5 \cdot 10^{-8}$ |



**Appendix F: Sea surface temperature**

Following Ward et al. (2018), the SST is limiting surface chlorophyll if it is higher than 20°C. Our results indicate that the averaged SST is limiting phytoplankton concentration for the simulations following the RCP8.5 scenario.

**Table F1.** Sea surface temperature (°C) for the different simulations.

| Simulations | SST(°C) |
|---|---|
| RCP2.6 | 15.95 |
| RCP2.6LA | 16.54 |
| RCP4.5 | 18.08 |
| RCP4.5LA | 18.68 |
| RCP6.0 | 19.18 |
| RCP6.0LA | 19.75 |
| RCP8.5 | 23.21 |
| RCP8.5LA | 23.44 |



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
