# Peer review of "A missing link in the carbon cycle: phytoplankton light absorption under RCP emissions scenarios"

_EGUsphere, 2023_

## Referee Comment (RC2)

**Review of "A missing link in the carbon cycle: phytoplankton light absorption under RCP scenarios" by Asselot et al.,**

**General Comments**

Asselot et al., quantify the impact of considering phytoplankton light absorption on the carbon cycle and climate under a multi-centennial future projections using an Earth System model. They find that resolving this additional process leads to increases in chlorophyll, sea surface and atmospheric temperatures, plus increases in atmospheric CO2.

The experimental design and key results appear to be robust. The justification for considering multi-centennial timescales and the light absorption could be clearer in the introduction though, with some useful context only appearing at the end of the manuscript. The explanation of why phytoplankton light absorption leads to the key is not currently robust – there are clarifications and more analysis needed on the role of temperature in the ecosystem model and, in my view, over interpretation of minimal biological carbon pump changes that needs addressing. One of the most interesting outcomes of the study for me is that the impact of light absorption has a non-linear dependence on the scenarios but unfortunately I don't think the explanation for this is as robust as it can be.

**Specific Comments**

Lines 20 – 35: Cael et al., (2023) is a recent addition to the literature on observations that should be cited.

Line 47: "Following RCP8.5 scenario" – may be better phrased as something like "Under a scenario of anthropogenic emissions,…" to better differentiate it from the 1% atmospheric CO2 increase experiment discussed in the previous experiment.

Line 63: "long timescale" – be more precise, do you mean centennial or millennial for example? Why is a >2100 timescale important to consider?

Section 2: This seems like a subsection of the Methods rather than its own individual section

Lines 83 – 84: the analysis is quantitative here in at least you quantify the net impacts. You don't quantify the components or drivers of those net impacts, but I don't think you need to frame that as qualitative!

Lines 132 -133: This could be more precise. For example, you can back out the percentage of POC remineralised from the e-folding depth (or the net flux from the curve if it's a double exponential) which gives a more intuitive metric here. This also needs to take into account the bottom depth of the euphotic layer which I think is different here than in Ward et al., (2018) so 590m may actually be a few hundred meters deeper. I am not sure where the <590m figure comes from DOM remineralisation as this is dependent on advection vs. remineralisation timescale – I think perhaps 590m gives the wrong impression of what's happening, maybe a more approximate number might help.

Lines 142 / 187 – 188 / Appendix B: What is the logic behind the choice of PFT cell sizes? Notably, the zooplankton size class is less than 10 times bigger than the phytoplankton which contrasts with the optimum grazing prey length ratio of 10 times smaller. This means the zooplankton type is not grazing optimally on the phytoplankton, e.g., the proportion of the prey biomass available to the grazer (eqn. 20 in Ward et al., 2018) equals 0.8471 here.

Line 142: "species" is not appropriate given the trait-based model, "group" or "type" might be better.

Lines 161 – 163 / 325 / Appendix A:
- I am struggling to see the suggested effect of SST on chlorophyll around 20 degrees C on Figure A1. Arguably, the upper part of the distribution of chlorophyll begins to decrease around 20 degrees but the lower part of the distribution decreases from 10 degrees.
- I don't think you can conclusively conclude on the relationship with SST because Figure A1 also includes other factors that may be co-varying with SST, e.g., nutrient availability. To do this, I think you'd need to plot this with a constant nutrient concentration or vary temperature whilst keeping nutrient concentrations fixed.
- The net effect of temperature dependence is quite complicated. Nutrient uptake and grazing *rates* increase with temperature, however net nutrient uptake can be limited by nutrient availability leading to disproportionate effects depending on location. For example, the temperature effect of grazing is more likely to dominate in areas with lower nutrient availability. This effect needs to be factored into the explanation of why the impacts under RCP8.5 are less pronounced.

Line 169: the six oceanic layers should appear in the ecosystem section as this is a departure from Ward et al., (2018)

Lines 190 – 191 / 349 – 350: I think the authors are correct in their assertion that the ecosystem will spin up rapidly with the initial biogeochemical state. However, the ecosystem will have an impact on the biogeochemistry via a different uptake of nutrients and carbon and because this impact is broadcast to the deep ocean via sinking particulates it's likely there is a much longer drift in the biogeochemistry. It would help to have an additional experiment to quantify this drift and its impact on the simulations. The alternative approach is to perform a second coupled biogeochemistry-ecosystem spin-up to allow the biogeochemistry to adjust.

Section 4.1.1:

- The variation in b values of around 0.01 reported in Table 1 is incredibly small given the observed spatial variability in the ocean (0.4 to 1.4: Henson et al., 2012; Marsay et al., 2015) and projected future values with temperature dependent remineralisation (~0.25; Laufkotter et al., 2017). The percentage of POC sinking beyond 1000m, an indication of carbon sequestration, calculated from a Martin Curve with the min/max values in Table 2 ranges from 20.8% to 21.4%. Overall, this suggests a very minimal change in the Biological Carbon Pump in response to the light absorption.
- "…we compute [vertical POC fluxes] via a Martin curve…" – I'm totally sure what this means, did you fit a power-law curve to the vertical profile of POC fluxes predicted by

the model? If so, what did you use as the normalisation depth and does this include POC generated in the upper 6 depth levels? Generally, this is not as straight forward as suggested because an exponential curve has linear attenuation whilst a power-law has non-linear attenuation (Lauderdale & Cael 2021).

- is the exponential decay function normalised to the bottom depth of the euphotic zone of the model (assuming this is the bottom of the sixth depth level where light penetrates)?
- The authors seem to suggest the change in remineralisation is occurring in the surface, where I assume the adjusting ecosystem is driving that change, rather than changing the attenuation of POC fluxes across the water column. It would help to see a vertical profile of POC fluxes to confirm this. If this is true and the changes in b reflect this, then this is slightly conflating concepts of POC attenuation, as measured by b, and changing export efficiency (the ratio of export at some reference depth to production: f-ratio, see Henson et al., 2011).

Line 229: "more labile" – this infers POC has different reactivity in the model, is this true?

Figure 4: It might help to have some indication of how big these changes are relatively, i.e., compared to the overall final-preindustrial change, though I appreciate the comparisons are focused on the final state with and without the light absorption.

Figure 6 and Section 4.1.3: The spatial patterns in SST differences for RCP8.5 look to be different to the other scenarios. There is greater warming at the poles compared to smaller warming in the other scenarios which is an interesting feature that doesn't seem to be discussed in the text.

Lines 286 – 289: It's not clear here whether the quoted changes in the carbon pumps is from the previous paper or this study.

Lines 375 – 378:

- It would help to give a sense of this change relative to the overall change in the carbon cycle to support your suggestion that phytoplankton light absorption leads to major carbon cycle uncertainties.
- "This study highlights a highly uncertain feedback on the carbon cycle that is missing from 50% of the CMIP6 models" – this is a crucial point for justifying this study which is left to the very end of the manuscript! This would be really beneficial to mention in the introduction.

**References**

Cael, B.B., Bisson, K., Boss, E. et al. (2023) Global climate-change trends detected in indicators of ocean ecology. *Nature* 619, 551–554. https://doi.org/10.1038/s41586-023-06321-z

Henson et al., (2011) A reduced estimate of the strength of the ocean's biological carbon pump. *Geophysical Research Letters.* 38, L04606. doi:10.1029/2011GL046735

Lauderdale, J. M., & Cael, B. B. (2021). Impact of remineralization profile shape on the air-sea carbon balance. *Geophysical Research Letters*, 48, e2020GL091746. https://doi. org/10.1029/2020GL091746

Laufkötter, C., J. G. John, C. A. Stock, and J. P. Dunne (2017), Temperature and oxygen dependence of the rem- ineralization of organic matter, *Global Biogeochemical Cycles*, 31, 1038–1050, doi:10.1002/2017GB005643.

Ward et al., (2018) EcoGEnIE 1.0: plankton ecology in the cGEnIE Earth system model, *Geoscientific Model Development.* 11, 4241–4267, https://doi.org/10.5194/gmd-11-4241-2018

---

## Author Comment (AC1)

Review of "A missing link in the carbon cycle: phytoplankton light absorption under RCP scenarios" by Asselot et al.,

General Comments

Asselot et al., quantify the impact of considering phytoplankton light absorption on the carbon cycle and climate under a multi-centennial future projections using an Earth System model. They find that resolving this additional process leads to increases in chlorophyll, sea surface and atmospheric temperatures, plus increases in atmospheric CO2.

The experimental design and key results appear to be robust. The justification for considering multi-centennial timescales and the light absorption could be clearer in the introduction though, with some useful context only appearing at the end of the manuscript. The explanation of why phytoplankton light absorption leads to the key is not currently robust – there are clarifications and more analysis needed on the role of temperature in the ecosystem model and, in my view, over interpretation of minimal biological carbon pump changes that needs addressing. One of the most interesting outcomes of the study for me is that the impact of light absorption has a non-linear dependence on the scenarios but unfortunately I don't think the explanation for this is as robust as it can be.

We would like to thank the reviewer for their very constructive comments, which greatly helped to improve the quality of the manuscript. Our responses are in blue, with edits to the manuscript in red.

Specific Comments

Lines 20 – 35: Cael et al., (2023) is a recent addition to the literature on observations that should be cited.

We thank the reviewer and added the sentence:
"For instance, satellite observations demonstrate that low-latitude oceans have become greener due to climate change between 2002–2022 (Cael et al., 2023)"

Line 47: "Following RCP8.5 scenario" – may be better phrased as something like "Under a scenario of anthropogenic emissions,…" to better differentiate it from the 1% atmospheric CO2 increase experiment discussed in the previous experiment.

We rephrased by:
"Additionally, the sensitivity of the light attenuation coefficient for phytoplankton is investigated under the RCP8.5 scenario (Kvale and Meissner, 2017)"

Line 63: "long timescale" – be more precise, do you mean centennial or millennial for example? Why is a >2100 timescale important to consider?

By "long timescale" we meant multi-century timescale. In section 2.6 of the revised manuscript we added:
"We consider a multi-century timescale to evaluate the long term influence of anthropogenic $CO_2$ emissions. Even if these emissions cease or are reduced by 2100, their influence will be echoed for centuries."

Section 2: This seems like a subsection of the Methods rather than its own individual section

We moved the "RCP" section into the Methods section.

Lines 83 – 84: the analysis is quantitative here in at least you quantify the net impacts. You don't quantify the components or drivers of those net impacts, but I don't think you need to frame that as qualitative!

We rephrase by:
"we focus on the quantification of the large-scale impacts of phytoplankton light absorption but we do not quantify the components or drivers of those large-scale impacts."

Lines 132 –133: This could be more precise. For example, you can back out the percentage of POC remineralised from the e-folding depth (or the net flux from the curve if it's a double exponential) which gives a more intuitive metric here. This also needs to take into account the bottom depth of the euphotic layer which I think is different here than in Ward et al., (2018) so 590m may actually be a few hundred meters deeper. I am not sure where the <590m figure comes from DOM remineralisation as this is dependent on advection vs. remineralisation timescale – I think perhaps 590m gives the wrong impression of what's happening, maybe a more approximate number might help.

We apology for the confusion here, the figure 590 m is wrong. In our study, the base of the euphotic layer is 221.84 m. The POM is predominantly remineralized below the base of the euphotic layer (221.84 m) while DOM is remineralized above this limit. Over the water column, 30% of the remineralization of POM occurs above the euphotic layer (0 – 221.84 m) while the remaining 70% occurs below this layer. We rephrase by:
"To achieve this, the flux is partitioned between POM, of which, on average, 70% is remineralized below the euphotic layer (0 – 221.84 m), and DOM which is predominantly remineralized within this layer."

Lines 142 / 187 – 188 / Appendix B: What is the logic behind the choice of PFT cell sizes? Notably, the zooplankton size class is less than 10 times bigger than the phytoplankton which contrasts with the optimum grazing prey length ratio of 10 times smaller. This means the zooplankton type is not grazing optimally on the phytoplankton, e.g., the proportion of the prey biomass available to the grazer (eqn. 20 in Ward et al., 2018) equals 0.8471 here.

In this study, we use the same PFT cell sizes as in our previous studies (Asselot et al., 2021; Asselot et al., 2022). In Asselot et al. (2021), we conducted a simulation with low ecosystem complexity (simulation A) and a simulation with a high ecosystem complexity (simulation B). The PFT cell sizes used in these simulations are tabulated below. The 12 PFT cell sizes of simulation B are a subset of the 16 sizes used by Ward et al. (2018), noting that we neglected the two largest phytoplankton and zooplankton classes because these were found to contribute negligible biomass. For consistency, the PFT cell sizes of simulation A were taken as the average class size of simulation B in preference to an idealised assumption of an optimal grazing prey length ratio. In Asselot et al. (2021) we showed that the climate impact of changing between these representations of ecosystem complexity is negligible compared to that from phytoplankton light absorption, and we here use only the low ecosystem complexity (PFT cell sizes corresponding to simulation A).
We have added the text:
"We consider only one phytoplankton and one zooplankton class size, following the low ecosystem complexity model of Asselot et al (2021), noting that Asselot et al (2021) found that the climate impact of changing ecosystem complexity was negligible compared to that from phytoplankton light absorption."

| | Simulation A | Simulation B |
|---|---|---|
| Phytoplankton (µm) | 46.25 | 0.60 |
| Phytoplankton (µm) | | 1.90 |
| Phytoplankton (µm) | | 6.0 |
| Phytoplankton (µm) | | 19.0 |
| Phytoplankton (µm) | | 60.0 |
| Phytoplankton (µm) | | 190.0 |
| Zooplankton (µm) | 146.15 | 1.90 |
| Zooplankton (µm) | | 6.0 |
| Zooplankton (µm) | | 19.0 |
| Zooplankton (µm) | | 60.0 |
| Zooplankton (µm) | | 190.0 |
| Zooplankton (µm) | | 600.0 |

Line 142: "species" is not appropriate given the trait-based model, "group" or "type" might be better.

We changed "species" to "group"

Lines 161 – 163 / 325 / Appendix A:
        - I am struggling to see the suggested effect of SST on chlorophyll around 20 degrees C on Figure A1. Arguably, the upper part of the distribution of

chlorophyll begins to decrease around 20 degrees but the lower part of the distribution decreases from 10 degrees.
- I don't think you can conclusively conclude on the relationship with SST because Figure A1 also includes other factors that may be co-varying with SST, e.g., nutrient availability. To do this, I think you'd need to plot this with a constant nutrient concentration or vary temperature whilst keeping nutrient concentrations fixed.
- The net effect of temperature dependence is quite complicated. Nutrient uptake and grazing rates increase with temperature, however net nutrient uptake can be limited by nutrient availability leading to disproportionate effects depending on location. For example, the temperature effect of grazing is more likely to dominate in areas with lower nutrient availability. This effect needs to be factored into the explanation of why the impacts under RCP8.5 are less pronounced.

Many thanks for these points, which we fully agree with and have addressed as follows.

In section 2.4 temperature dependence, we added:
"Photosynthesis is light limited, which results in a sub-exponential growth rate, while competing effects of nutrient demand and zooplankton predation increase exponentially and together progressively limit net productivity as temperatures increase. We note that temperature dependence may be complicated by co-varying factors such as nutrient availability, leading to disproportionate effects depending on location. To explore these dependencies, chlorophyll and nutrient density are plotted against SST in appendices A1 and A2 respectively, with data separated into binned subsets with different nutrient density. When nutrient density is low (< 0.017 mmol/m3), 30% of the variance in chlorophyll is explained by temperature, with a negligible contribution of co-varying nutrient (only 7% of nutrient variance can be explained by SST in this bin). In contrast, under high nutrient concentrations (>0.1 mmol/m3), while 51% of the variance in chlorophyll can be explained by temperature, as much as 41% of this could be explained by co-variance of nutrients with temperature. In summary chlorophyll is limited by increasing temperature both through increased nutrient demand and zooplankton grazing, and through reduced nutrient availability, likely, at least in part, driven by the increasing nutrient demand."

In 4.1 General discussion
"However, under the RCP8.5 scenario, the effect of phytoplankton light absorption on the climate system is reduced. This is likely due to decreasing ecosystem productivity as temperature increases (Appendix A1 and A2), caused by exponentially increasing nutrient demand and zooplankton predation, combined with sub-exponential (light limited) increases in photosynthesis."

Line 169: the six oceanic layers should appear in the ecosystem section as this is a departure from Ward et al., (2018)

We added this information in the "Ecosystem community component" section. See comment line 132–133.

Lines 190 – 191 / 349 – 350: I think the authors are correct in their assertion that the ecosystem will spin up rapidly with the initial biogeochemical state. However, the ecosystem will have an impact on the biogeochemistry via a different uptake of nutrients and carbon and because this impact is broadcast to the deep ocean via sinking particulates it's likely there is a much longer drift in the biogeochemistry. It would help to have an additional experiment to quantify this drift and its impact on the simulations. The alternative approach is to perform a second coupled biogeochemistry-ecosystem spin-up to allow the biogeochemistry to adjust.

We agree that switching on ECOGEM will have an impact on the biogeochemistry. However, our results are focused upon the impact of light absorption relative to simulations without light absorption, so the drift is explicitly accounted for - it is common to the experiments with and without the effect. We have clarified this with the text:
"Switching on ECOGEM has an impact on the biogeochemistry via a different uptake of nutrients and carbon. However, we are interested in the effect of light absorption by phytoplankton relative to simulations without light absorption and our experimental results are differences between two otherwise identical simulations; the altered atmospheric CO2 and subsequent long-term drift in the carbon cycle induced by ECOGEM are common to both experiments."

Section 4.1.1:
– The variation in b values of around 0.01 reported in Table 1 is incredibly small given the observed spatial variability in the ocean (0.4 to 1.4: Henson et al., 2012; Marsay et al., 2015) and projected future values with temperature dependent remineralisation (~0.25; Laufkotter et al., 2017). The percentage of POC sinking beyond 1000m, an indication of carbon sequestration, calculated from a Martin Curve with the min/max values in Table 2 ranges from 20.8% to 21.4%. Overall, this suggests a very minimal change in the Biological Carbon Pump in response to the light absorption.

To compare the strength of the biological carbon pump, we changed our approach. In our model setup, the POC exponential is fixed and spatially invariant so it wasn't surprising the b values were roughly constant when we fit to the data. In the revised manuscript we compare the global POC flux between simulations, which defined the amount of POC transported to the deep ocean. In section 3.1.1 we modified Table 2 and add the text:
"To compare the strength of the biological carbon pump between our simulations, we consider vertical fluxes of POC in the water column. In our study, these fluxes are described by an exponential decay, which is fixed and spatially invariant. Under RCP2.6, RCP4.5 and RCP6.0 scenarios, the POC flux decreases by 4–5% when phytoplankton light absorption is simulated (Table 2). For the RCP8.5 scenario, the

effect is smaller, with a POC flux reduced by 1% due to phytoplankton light absorption. In our simulations, independently of the RCP scenario, phytoplankton light absorption decreases the POC flux (Table 2), indicating that less organic matter is transported towards the bottom of the ocean. This reduced export efficiency is due to an enhanced remineralization at the ocean surface, which is driven by a higher amount of organic matter in the ocean surface. Indeed, the surface net primary production increases with phytoplankton light absorption (Table 2), leading to an enhanced remineralization at the ocean surface. These results indicate that biological pump is weaker with phytoplankton light absorption meaning that more inorganic matter, such as nutrients, is located in the surface of the ocean (Table E1)."

- "…we compute [vertical POC fluxes] via a Martin curve…" – I'm totally sure what this means, did you fit a power-law curve to the vertical profile of POC fluxes predicted by the model? If so, what did you use as the normalisation depth and does this include POC generated in the upper 6 depth levels? Generally, this is not as straight forward as suggested because an exponential curve has linear attenuation whilst a power-law has non-linear attenuation (Lauderdale & Cael 2021).

We modified our approach and we use the POC flux as a proxy to study the strength of the biological carbon pump (see above). We do not fit our data to a Martin curve anymore.

- is the exponential decay function normalised to the bottom depth of the euphotic zone of the model (assuming this is the bottom of the sixth depth level where light penetrates)?

The exponential decay function is normalized to the third oceanic layer (78 m), where the POC flux is maximum.

- The authors seem to suggest the change in remineralisation is occurring in the surface, where I assume the adjusting ecosystem is driving that change, rather than changing the attenuation of POC fluxes across the water column. It would help to see a vertical profile of POC fluxes to confirm this. If this is true and the changes in b reflect this, then this is slightly conflating concepts of POC attenuation, as measured by b, and changing export efficiency (the ratio of export at some reference depth to production: f-ratio, see Henson et al., 2011).

We thank the reviewer for this point. Indeed, the adjusting ecosystem drives change in remineralization and changes in attenuation of POC fluxes. To prove the latter, we computed the f-ratio and ThEi-ratio (see table below). These two parameters define the fraction of organic matter exported in the deep ocean (Henson et al., 2011). A decrease in these two parameters indicates a reduced export efficiency of organic matter. Here, these two parameters are computed via the global SST of each simulation, as defined in Henson et al. (2011). Independently of the RCP scenario considered, phytoplankton light absorption decreases the f-ration and ThEi-ration,

indicating that this biogeophysical mechanism reduced the export efficiency of organic matter and thus weakens the biological pump.
In the revised manuscript, we changed our approach and rephrased the whole section (see above).

| Simulation | f-ratio | ThEi-ratio |
| --- | --- | --- |
| RCP2.6 | 0.3010 | 0.0642 |
| RCP2.6LA | 0.2892 | 0.0612 |
| RCP4.5 | 0.2584 | 0.0541 |
| RCP4.5LA | 0.2464 | 0.0516 |
| RCP6.0 | 0.2364 | 0.0496 |
| RCP6.0LA | 0.2250 | 0.0474 |
| RCP8.5 | 0.1558 | 0.0359 |
| RCP8.5LA | 0.1512 | 0.0353 |

Line 229: "more labile" – this infers POC has different reactivity in the model, is this true?

We apologize for the confusion here. By "labile organic matter" we meant dissolved organic matter. We rephrased by:
"This increase is due to the increased global phosphate concentrations (Appendix D1) which are driven by a reduced export efficiency of organic matter and enhanced remineralization at the ocean surface (Table 2)"

Figure 4: It might help to have some indication of how big these changes are relatively, i.e., compared to the overall final-preindustrial change, though I appreciate the comparisons are focused on the final state with and without the light absorption.

As stated by the reviewer, the scope of this study is to compare the state of the climate system with and without phytoplankton light absorption (PLA). As an indication, we computed the changes in chlorophyll compared to the pre-industrial era but we do not report these values in the revised manuscript. The second column of the table below represents the changes between the simulations with minus without phytoplankton light absorption (values on Fig. 4). The third column represents the difference in chlorophyll between our simulations with phytoplankton light absorption minus the pre-industrial state. Our results indicate that, following future climate scenarios, chlorophyll decreases.

| Scenario | PLA – NoPLA | PLA - PreInd |
| --- | --- | --- |
| RCP2.6 | +13% | -4% |
| RCP4.5 | +12% | -5% |
| RCP6.0 | +15% | -3% |
| RCP8.5 | +8% | -7% |

Figure 6 and Section 4.1.3: The spatial patterns in SST differences for RCP8.5 look to be different to the other scenarios. There is greater warming at the poles compared

to smaller warming in the other scenarios which is an interesting feature that doesn't seem to be discussed in the text.

Thank you for this point, we added the explanation:
"In contrast, under RCP8.5, the maximum SST increase of 0.51°C occurs in the Southern Ocean. This is due to the greatly reduced annually averaged sea ice under RCP8.5, meaning that the latent heat buffering effect of melting/growing sea-ice is weaker, allowing heating of the ocean surface. The annual ice cover in the simulation RCP8.5-LA is only $5.1 \times 10^6$ km$^2$ of the global ocean surface at 2500, which compares to $25.8 \times 10^6$ km$^2$ for RCP2.6-LA."

Lines 286 – 289: It's not clear here whether the quoted changes in the carbon pumps is from the previous paper or this study.

These results are actually from this study. We rephrased by:
"Our results indicate that the reduced solubility pump…"

Lines 375 – 378:
- It would help to give a sense of this change relative to the overall change in the carbon cycle to support your suggestion that phytoplankton light absorption leads to major carbon cycle uncertainties.

Implementing phytoplankton light absorption increases the atmospheric carbon content by ~23% and ~8% under RCP2.6 and RCP8.5 scenario, respectively. We rephrased by:
"For instance, with our model setup, implementing phytoplankton light absorption increases the atmospheric carbon content by 79 GtC (23%) under RCP2.6 and by 258 GtC (8%) under RCP8.5, compared to the simulations without this biogeophysical mechanism."

- "This study highights a highly uncertain feedback on the carbon cycle that is missing from 50% of the CMIP6 models" – this is a crucial point for justifying this study which is left to the very end of the manuscript! This would be really beneficial to mention in the introduction.

At the end of the second paragraph of the introduction we added the sentence:
"All these previous studies have demonstrated that phytoplankton light absorption affects the future climate projections but, to this day, this biogeophysical mechanism is missing from 50% of the CMIP6 models (Pellerin et al., 2020)."

References

Cael, B.B., Bisson, K., Boss, E. et al. (2023) Global climate-change trends detected in indicators of ocean ecology. Nature 619, 551–554. https://doi.org/10.1038/s41586-023-06321-z

Henson et al., (2011) A reduced estimate of the strength of the ocean's biological carbon pump. Geophysical Research Letters. 38, L04606. doi:10.1029/2011GL046735

Lauderdale, J. M., & Cael, B. B. (2021). Impact of remineralization profile shape on the air-sea carbon balance. Geophysical Research Letters, 48, e2020GL091746. https://doi.org/10.1029/2020GL091746

Laufkötter, C., J. G. John, C. A. Stock, and J. P. Dunne (2017), Temperature and oxygen dependence of the rem- ineralization of organic matter, Global Biogeochemical Cycles, 31, 1038–1050, doi:10.1002/2017GB005643.

Ward et al., (2018), EcoGEnIE 1.0: plankton ecology in the cGEnIE Earth system model, Geoscientific Model Development. 11, 4241–4267, https://doi.org/10.5194/gmd-114241-2018

Asselot, R., et al. (2021) "The relative importance of phytoplankton light absorption and ecosystem complexity in an Earth system model". Journal of Advances in Modeling Earth Systems 13.5: e2020MS002110.

Asselot, R., et al. (2022) "Climate pathways behind phytoplankton-induced atmospheric warming." Biogeosciences 19.1: 223–239.

Pellerin, F. (2020) "ESD Reviews: Evidence of multiple inconsistencies between representations of terrestrial and marine ecosystems in Earth System Models." Earth System Dynamics Discussions: 1–26.

Ward, B., et al. (2018), EcoGEnIE 1.0: plankton ecology in the cGEnIE Earth system model, Geoscientific Model Development. 11, 4241–4267, https://doi.org/10.5194/gmd-114241-2018

---

## Author Comment (AC2)

Review of the paper "A missing link in the carbon cycle: phytoplankton light absorption under RCP scenarios"

General comments

I have completed the review of the manuscript egusphere-2023-921. The manuscript shows some interesting results, somehow scientifically soundings. At the same time, I found that the manuscript has several issues that the Authors should address before a possible publication of their work. Thus, I asked major revisions for this work.

We would like to thank the reviewer for their very interesting comments. The review helped to improve the quality of the manuscript and the science behind it. Our responses are in blue, with edits to the manuscript in red.

Major comments

1. The manuscript shows several typos. See for example "phytopankton" at line 8. Please consider a strong editing of text.

We used an online editing tool to remove the typos in the revised manuscript.

2. There is a lack in explaining the purpose and the methodology followed in the study.

At the end of the introduction we added the text to clarify:
"The purpose of this study is to better understand how phytoplankton light absorption will be affected by anthropogenic climate change via changes in phytoplankton biomass and distribution. To address this question, we performed simulations with and without phytoplankton light absorption in experiments with prescribed atmospheric $CO_2$ emissions. We are interested in long-term climate effects and so we applied the intermediate complexity Earth system model EcoGEnIE (Ward et al., 2018). We force the model with atmospheric $CO_2$ emissions out to 2500 following the four Extended Representative Concentration Pathways (RCP) scenarios used by the Intergovernmental Panel on Climate Change (IPCC) for their Fifth Assessment Report (Moss et al., 2010)."

3. Along the text the Authors mention the "primary production". Do you mean "net primary production" (gross pp minus respiration) or the gross pp? please specify since they are different things that can lead to a different interpretation of your outcomes.

In the manuscript we referred to net primary production. We revised the manuscript accordingly.

4. There is a lack in the manuscript in the description of the modeling tool and forcing adopted in the simulations. For example, I do not understand why using such old values for wind forcing (Trenberth; 1989) since you run long

term simulations covered several centuries, eventually covered by CMIP models.

We apply the model with an EMBM atmosphere for computational efficiency and simplicity. The wind forcing of Trenberth et al. (1989) is the default wind forcing in this configuration of EcoGEnIE. The EMBM is a single layer diffusive model and is largely insensitive to the details of the wind field forcing, so that there would be little benefit in applying e.g. time varying wind fields derived from high complexity models. A full description of the model is beyond the scope of this paper but it is described in detail in Ward et al (2018) and references therein.

5. I'm puzzled about the fact that chl-a is not transported by ocean currents. Is it your model result? Is it an assumption of the analysis or a constrain in your numerical simulations? Please explain better. Phytoplankton (and chl-a) could be considered a passive tracer (as it is in many coupled models). Thus, in marine environment it can transported by advection or diffusion processes.

We clarify with the following addition in the section "Ecosystem community component":
"Living matter is not subject to ocean transport. Communication between biological communities only occurs through the advection and diffusion of inorganic and non-living organic matter. This approximation is justified by the coarse (~1000 km) model resolution and limited transport range of living matter, so the rate of transport between grid cells is slow in relation to the net growth rates of the plankton community (Ward et al 2018)."

6. The explanation provided by the Authors related to the increase in the vertical velocity (see also my comment on Appendix D) looks to me weird (maybe I'm missing something in their reasoning and so please help me to understand). If someone warms/cools the upper/bottom of the water column I would expect an increase of the vertical stratification and thus lower vertical transport and vertical mixing (as predicted by several studies discussing the future climate projections). The Authors talk about an increase in the vertical velocity because of the difference in temperature. I do not think this is correct. Please explain better this point.

We thank the reviewer for this interesting discussion. We agree that a warming of the surface ocean would drive an increase of the stratification and that our arguments to the contrary were not robust. We note that there is a suggestion of increased upwelling in the Pacific coast of the Americas and the Atlantic coast of Africa. However, these changes are rather modest and noisy. In view of this we have decided to remove mention of this potential mechanism. Reduced export efficiency of organic matter and enhanced remineralization at the ocean surface are sufficient to explain the patterns of chlorophyll change, and we have restricted our argument to this mechanism. For the reviewer's interest, we provide 2D maps of change in vertical velocity driven by phytoplankton light absorption in the RCPs, illustrating the increased (but weak) upwelling signal along the Pacific coast of the Americas and the Atlantic coast of Africa.

[Figure]

Specific comments

Line 2: I would say influence not impact.

Changed

Line 8–12: Please explain better how the light absorption would weaken the carbon pump.

We rephrased to:
"Under all RCP scenarios, our results indicate that phytoplankton light absorption leads to a shallower remineralization of organic matter and a reduced export efficiency, weakening the biological carbon pump."

Line 16–19: This part is not very clear and should go at the end of the introduction as purpose of the work.

We removed these two sentences at the beginning of the introduction and had one sentence at the end of the introduction to better explain the purpose of this work. The sentence added is:
"The purpose of this study is to better understand how phytoplankton light absorption will be affected by anthropogenic climate change via changes in phytoplankton biomass and distribution."

Line 25–35. There is a dependence of the projections for the net primary production on the parametrization adopted and how they are influenced by the temperature. In

the Mediterranean sea there is an extensive review on this topic by Richon et al., 2019 and Reale et al., 2022. Please consider to add a sentence about.

We thank the reviewer for pointing out these two interesting studies. We add three sentences in the introduction:
"On a regional scale, projected changes in primary production are also uncertain. For instance, in the Mediterranean Sea, Richon et al. (2019) show a decline in net primary production of 10% in the 2090s under the high-emission SRES-A2 scenario. However, in the same basin, Reale et al. (2022) demonstrate that, under the RCP4.5 and RCP8.5 scenarios, the net primary production increase is greater than 10 gC/m2/yr by the end of the 21st century. These conflicting results come from the different parameterizations adopted which exert differing influences of temperatures on simulated net primary production."

Line 33–35: how? Please explain

In the revised version of the manuscript, we removed this sentence and replaced it by the sentences describing the project changes in net primary production in the Mediterranean Sea (see comment above).

Line 38 and line 55: On what? Please explain

We meant on the oceanic temperature. We revised the sentences accordingly.

Line 68 e Fig.2: The RCP scenarios do not "describe possible future climate system" but hypothetical temporal evolution of greenhouse gases emission in the atmosphere. Please correct this statement and the caption.

We rephrase line 68 by:
"The RCP scenarios include the temporal evolution of greenhouse gas emissions into the atmosphere (Moss et al., 2010)".
We rephrase the caption of Fig. 2 by:
"Atmospheric CO2 emissions following the RCP scenarios. (a) Historical and scenarios of future CO2 emissions over time (GtC/yr)."

Line 81–82-89: What do you mean with "association", "intermediate" and "related to climate processes"?

With "association" we mean that EcoGEnIE is a union/coupling between the new ECOGEM component and the previous model cGEnIE. We rephrase.
According to Claussen et al. (2002), ESM of intermediate complexity are designed to represent the Earth system, excluding the interactions between humans and nature. These models were created to close the gap between complex coupled general circulation models (CGCMs; designed to represent as much climate feedbacks as possible and computationally expensive) and simplistic models (designed to study the plausibility of climate processes and often represent only one component of the climate system).

We rephrase and replace "related to climate process" by "represent climate processes".

Figure: is there a coupler managing the exchange of fields (arrows) among the components of the modeling tools? Please specify

The coupling is now summarised in the caption of Fig. 1
"GEnIE is controlled by a bespoke coupling manager which was developed for user-friendly modularity and flexibility, so that, for instance the EMBM atmosphere can be replaced with a fully dynamic 3D atmosphere PLASIM (Holden et al 2016) via a single switch in the model configuration file."

Line 101: I see that the purpose of the work is not to validate your simulations but please add some quantitative information to your discussion (under/over estimation is too generic).

We modify the text as follow:
"In contrast, Ridgwell et al. (2007) indicate that the low-latitude upwelling in the Western Equatorial Pacific and Equatorial Indian Ocean give an excess of phosphate of 0.5 µmol/kg compared to observations (Conkright and Levitus, 2002)."

Line 123: Why do not you consider Nitrate? Please specify.

We do not consider nitrate here because our model does not have an explicit representation of the nitrate cycle, but represent it through the Redfield ratio. We add the sentence:
"Similar to Asselot et al. (2021), we do not explicitly consider nitrate (NO3) but approximate it through the N:P Redfield ratio of 16:1 (Ridgwell 2007)."

Line 145: DIC as nutrient sounds to me weird since it involves different chemical species ($CO_2aq$, $HCO_3$, $CO_3$ and so on). Could you please add some references or explain better this point?

We thank the reviewer for this point. We changed "nutrients" to "inorganic resources".

Line 169: What are the criteria to choose the sixth layer as limit for light?

We chose the sixth layer as limit for light because it represents the base of the euphotic or sunlight zone (about 200 meters). We add this explanation in the manuscript.

Line 196: Why just one year (2050) instead of the entire period?

The scope of this study is to understand the long-term effect of phytoplankton light absorption on the climate system under long-term anthropogenic CO2 emissions. The climate system needs to adjust to these emissions, thus we compare only the year 2500, when the climate system has mostly responded to the CO2 forcing.

Studying the whole period would have implied studying transient effects, which is not the aim of our study.

Line 200: It would be nice to see the spatial distribution of these differences that could explain the differences among the different scenarios instead of a single value that is meaningless

We agree that it would be more informative to have the spatial distribution of these differences but Zickfeld et al. (2013) do not give any map showing these differences. The authors only give the global mean values of these differences. We note there are substantial uncertainties associated with the global warming response, and, as is appropriate for a model of intermediate complexity, we validate the model's climate carbon-cycle response to CO2 emissions using this large-scale metric.

Line 226: I do not think that chl-a is a climate variable

We replaced "climate variable" by "climate carbon-cycle variable".

Line 229: How? Please explain

We rephrase by:
"This increase is due to the increased global phosphate concentrations (Appendix D1) which are driven by a reduced export efficiency of organic matter and enhanced remineralization at the ocean surface (Table 2)."

Line 230-235: see my major point 6

We refer the reviewer to our answer to major point 6. We removed the argumentation with the enhanced vertical velocity.

Line 266: what do you mean with underestimation of the oceanic circulation?

We apologize for the confusion here and changed the argumentation by:
"The polar regions experience the lowest changes in SST because temperatures are buffered by latent heat through melting sea-ice and remain close to freezing."

Line 269: What do you mean "The missing….model setup". Please explain

For clarification, we combined the two sentences and rephrase by:
"The differing spatial patterns between chlorophyll and SST can be explained by the fact that short-lived chlorophyll is not subject to transport, while (conserved) physical quantities, such as heat, are transported by oceanic currents."

Line 274-276: Please rephrase.

We rephrase by:
"The atmospheric CO2 concentrations in our simulations do not match the atmospheric concentrations of Meinshausen et al. (2011) in 2500. This is because our version of the model, with light penetrating until the sixth oceanic layer, has been

tuned to get reasonable net primary production and nutrient fields but not to get future atmospheric CO2 concentrations"

Line 342-342: Not clear. Please rephrase

We rephrase by:
"Our model setup allow for light and primary production until the sixth oceanic layer and this configuration has not been tuned to match projected atmospheric CO2 concentrations, leading to an underestimation of the latter."

Appendix D: It would be better to have a map to show the global distribution of this quantity. Maybe the results in the Chilean area could be associated with La Nina/El nino pattern. Did you check that?

We removed this argument on vertical upwelling as detailed above, but have included the suggested figure in this response for the reviewer's interest.

---

## Referee Report (RR1)

Review of **"A missing link in the carbon cycle: phytoplankton light absorption under RCP emissions scenarios"**
by Rémy Asselot, Philip Holden, Frank Lunkeit and Inga Hense

**General comments**

Following the conclusions of Asselot et al (2022) who demonstrate that phytoplankton light absorption (PLA) mainly affects the climate system via air-sea CO2 exchange, the present study of Asselot and co-authors analyse the effect of activating the PLA in an earth system model (ESM) of intermediate complexity under emissions-driven (for CO2) scenarii of climate change.

Thanks to their framework with freely-evolving atmospheric CO2 concentrations, the authors show that the consideration of the PLA is critical, as it leads to an enhanced greenhouse gas effect in climate forecasts. Indeed it increases by 8 to 20% the global atmospheric CO2 concentrations. This result has great implications for climate forecasts: it highlights the importance of PLA-induced climate changes that a large proportion of current ESM do not consider, and proposes a quantification of this missing part of the atmospheric CO2 content under different climate scenarii (it identifies the PLA-induced changes as a function of climate change itself).

However, two points need clarifications from my perspective:
        - What implications has the use of an EMIC compared to a classical ESM in terms of feedbacks between the ocean and atmosphere ? This would give some clues on how your main results are generalizable.
        - One of the main result of this study is that PLA increases the surface net primary production in mid-latitude and upwelling regions due to a higher availability of nutrient concentrations, which is in turn driven by a higher remineralization at the ocean surface (and a reduced export efficiency). While this result is important, I would expect to understand to what perturbations of the oceanic physical conditions the higher remineralization is due.

**Specific comments by section**

**Abstract**

**L.9** "This biogeophysical mechanism increases the surface chlorophyll": based on your Table 2, your net primary production (globally integrated) in 2500 increases by less than 2% for all RCPs when activating the PLA. Of course this is not comparable with the order of changes you cited in your introduction part (e.g. "chlorophyll concentration has declined over more than 62% of the ocean surface from 1890 to 2010", "between 1998 and 2006, low surface chlorophyll areas have expanded by 15%"...), but I would highlight that point in the abstract by giving the percentages of changes, because your results show that, by triggering NPP changes of less than 2%, the PLA may perturb the global atmospheric CO2 content by 8 to 20%.

**L.15** "that *may be"* or "that *are maybe"* ?

**2 Methods**

From the **legend of figure 1**, I understood that what differentiates your EMIC from what you call an "ESM of high complexity" here is mainly the use of a simplified atmospheric module ("EMBM") which is not a fully 3D atmospheric model…? Please, could you clarify that aspect in the text of section 2, and explain with one sentence what is EMBM: if not a 3D model, is it a slab layer of atmosphere ?

**l.72-74** the authors wrote "EcoGEnIE is an ESM of intermediate complexity (EMIC) (Claussen et al., 2002) and due to the limitations of such a model, we focus on the quantification of the large-scale impacts of phytoplankton light absorption but we do not quantify the components or drivers of those large-scale impacts".

I am wondering how could we trust the large-scale impacts of PLA analyzed here if we do not trust what cause them ? I understand from this sentence that, due to the limitations inherent to an EMIC, the authors do not trust the drivers of the PLA large-scale impacts. Please reformulate.

**l.75-77** "We chose to conduct our study with an EMIC because we are interested on the effect on particular climate mechanism (e.g. phytoplankton light absorption) and it would have been difficult to isolate this effect with an ESM of high complexity, due to numerous climate feedbacks implemented in high complexity ESM."

In their analysis of many ESM "of high complexity", Séférian et al (2020) decomposed the Earth system interactions represented in ESM involving marine biogeochemistry into 4 main feedbacks: climate-carbon cycle feedbacks (F1), biogenic aerosol-cloud feedbacks (F2), non-CO2 biogeochemical cycle feedbacks (F3) and phytoplankton-light feedbacks (F4). It is not straithforward to me to see how climate feedbacks F1 to F3 would have perturb your analyses of the PLA-induced effects. Please, be more specific : give examples of the numerous feedbacks that would hinder the identification of PLA-induced effects. Don't you mostly think here to ocean-atmosphere interactions (not existing in your case due to the use of a simplified atmosphere with EMBM) ? If true, please mention it.

Séférian, R., Berthet, S., Yool, A. *et al.* Tracking Improvement in Simulated Marine Biogeochemistry Between CMIP5 and CMIP6. *Curr Clim Change Rep* **6**, 95–119 (2020). https://doi.org/10.1007/s40641-020-00160-0

**2.1 Ocean, atmosphere and sea-ice representation**

**l.95** "However, on a global scale, Marsh et al. (2011) show that the model simulates realistic upwelling."

With an horizontal resolution smaller than 3° in latitude (and not specified in longitude... but Ward et al., 2018 declare that they have 10° of longitudinal increments: what about yours ?…) and a minimum vertical spacing of 29 m, I guess "upwelling" refers to equatorial convergence, and not to coastal upwelling regions which have widths < 100 km, associated to very specific coastal dynamics needing quite fine horizontal and vertical resolutions to be represented. But even for equatorial regions, I find a bit inappropriate the expression "the model simulates realistic upwelling" as we know that your model represents only the very large-scale ocean dynamics. Could you describe in the text the dynamical conditions favoring these "realistic" upwellings in your model ?

**l.96-107** Again, could you clarify why this 2D atmospheric model was a more suitable choice than the fully 3D atmospheric model PLASIM in your framework ? Could you add a sentence explaining how the use of this simplified atmosphere may help revealing the PLA effects ?

**2.3 Ecosystem community component**

**l.126** "messy feeding" ?
**l.145** "so the rate"

**2.8 Model inter-comparison**

**l.225** comparison with an other EMIC ("an ESM of intermediate complexity") model: but more generally the reader is curious to know what would give the comparison with a high-complexity ESM ?

More generally, I understand that this first comparison focused on surface atmospheric temperature (SAT) because it allows to validate the use of a simplified atmospheric model in this study...? But back to the main goal of this study (effect of the PLA), I would expect here some elements characterizing how the ocean compartment absorbs heat without PLA (ocean heat content or at least ocean temperature). This would allow to discuss later the true PLA effect added by your equation (3). If Zickfeld et al (2013) have no ocean heat data, I suggest you to insert a small paragraph (and figure) characterizing the ocean heat content changes (or time series: see for example Figure 1 of Berthet et al, 2023) for each of your RCPs without and with PLA. Based on eq. (3), the first effect PLA will have on climate before any feedbacks on the biological pump/CHL/atmospheric CO2/SAT, will be to perturb the oceanic temperature, no ? So the first question for me is: how much ocean heat content is altered by the activation of your PLA parameterization ? Could you elaborate a bit on that point ?

Berthet, S., Jouanno, J., Séférian, R., Gehlen, M., and Llovel, W.: How does the phytoplankton–light feedback affect the marine $N_2O$ inventory?, Earth Syst. Dynam., 14, 399–412, https://doi.org/10.5194/esd-14-399-2023, 2023.

**3 Results**
**3.1 Oceanic properties**

**l.240** I would suggest to reformulate: "when PLA is activated" or "represented", rather than "simulated" ; this is only a suggestion, as my english is for from being perfect.

**L.241** How is the spatial pattern of this POC flux reduction ? Do you observe a reduction over the entire globe ? Or is it consistent with the patterns you described for the chlorophyll (l.266-270), i.e. mainly marked in upwelling and mid-latitude regions ?

**L.242** "independently of the RCP scenario": by activating the PLA, the oceanic temperature increases in sub-surface in all scenarii, but with different intensities, no ?

**l.245** "Indeed, the surface net primary production increases with phytoplankton light absorption": could you explain why ? To what perturbations of the physical conditions is it due (see my general comment on section 2.8) ? Please elaborate on that.

**3.1.2 Surface chlorophyll**

**L.266** Unlike the results of Paulsen (2018), who reports a decline in chlorophyll concentrations in the upwelling regions with PLA (**L.46**), you find a higher chlorophyll (CHL) concentration in the upwelling and mid-latitude regions with PLA in your model and framework: could you explain why Paulsen obtained an opposite feedback with its "Earth system model of high complexity" ? By which mechanism ? Does your EMIC represent this mechanism ? Or is this different behaviour attributable to the fact that Paulsen run its ESM under prescribed future atmospheric CO2 concentrations rather than freely-evolving emissions: in this case could you explain by which mechanism the atmospheric CO2 concentration may constrain the CHL to decrease locally in upwelling regions ?

**3.1.3 Sea surface temperature**

**L.273** "Due to changes in surface chlorophyll, we expect variations in SST".
What do you mean exactly here ? Due to changes in 1) surface chlorophyll *concentration* or 2) *in absorption properties* of surface chlorophyll ? This does not imply the same chain of causality:

–	case 1) describes the fact that PLA activation directly affects CHL concentration and, then, indirectly affects the SST due to the CHL concentration changes. However, in this case, could you clarify what mechanism triggers the initial perturbation of your CHL concentration ? In other words, how the PLA activation affects your CHL concentration ?

–	case 2) describes the fact that activating PLA has first a direct effect on ocean temperature. And that the other effects on CHL/export/remineralization arise from that one.

**3.2.1 Atmospheric CO2 concentration**

**L.293** I am a bit puzzled about these runs driven by CO2-emissions that do not match the target. I am not sure to fully understand the implications that could have on your analyses. Could you elaborate on it ?

**L.302** "For the RCP8.5 scenario, the atmospheric CO2 concentration increases by 8% only, which is due to the lower increase in chlorophyll and SST".

To demonstrate this assertion it would be interesting to see maps of atmospheric and oceanic CO2 partial pressure, as well as DpCO2 for all RCPs. Because in the current state I am not sure you have enough elements to directly conclude what you wrote. For me, you need to disentangle here 1) how the PLA activation changes your ocean CO2 content, from 2) how the CO2 atmospheric content in RCP8.5 allows to absorb new oceanic outgassing compared to the other RCPs. Your results in figures 7 and 8 show a non-linear behaviour of RCP8.5 compared to the other three RCPs, which is most likely attributable to a non-linear effect of the increased atmospheric CO2 concentrations.

From my perspective you did not dig deep enough into this aspect, because it was one of the main conclusions of your work: the effect of PLA activation is not linear and depends on the climate scenario. You may show that RCP8.5 crossed a tipping point (due to an extensive ice melt and other effects) what changes the way the ocean-atmosphere system manages the CO2 exchanges and finally, modulates the effect/amplitude the PLA activation may have on climate.

**4.1 General discussion**

**L.334** "Our results show that phytoplankton light absorption affects water temperature and nutrient concentrations."
Please see my related comment in section 3.1.3.

**L.335** "The increase in surface nutrient concentrations (Appendix D1) is driven by a reduced export efficiency of organic matter and enhanced remineralization at the ocean surface (Table 2)."
While I found the result of more remineralization in surface very interesting, I still have the feeling that something is missing in your analyses. You did not explain (or I missed it, so maybe it would be great to clarify it) by what mecanisms does the PLA activation affect your modelled remineralization and export ? This will possibly also help understand why RCP8.5 does not react proportionnally to the other RCPs when activating the PLA.

**L.336** "The increased surface nutrient concentrations leads to higher surface chlorophyll, which in turn leads to a warming of the ocean surface."
Here I interpreted that you choose the case (1) of my comment in section 3.1.3.

**4.2 Limitations**

**L. 384** "Our results highlight that phytoplankton light absorption itself increases chlorophyll leading to more heat being trapped in the ocean surface."
My guess is that PLA *promotes environmental conditions in ocean surface temperature* that allow an increase in remineralization in surface, what triggers an increase in nutrients concentrations and, thus,

an increase in CHL concentrations allowing more heat to be trapped: is that what you mean ? Please clarify.

---

## Author Response (AR2)

**First reviewer**

I have completed the review of the manuscript and the replies to my comments by Asselot et al. I would like to thank the Authors for taking into account my comments. The manuscript is now more clear and scientifically soundings. I'm happy to suggest its publication in present form.

We thank the reviewer for its comments that improve the science behind our article.

Review of "**A missing link in the carbon cycle: phytoplankton light absorption under RCP emissions scenarios**"
by Rémy Asselot, Philip Holden, Frank Lunkeit and Inga Hense

**General comments**

Following the conclusions of Asselot et al (2022) who demonstrate that phytoplankton light absorption (PLA) mainly affects the climate system via air-sea CO2 exchange, the present study of Asselot and co-authors analyse the effect of activating the PLA in an earth system model (ESM) of intermediate complexity under emissions-driven (for CO2) scenarii of climate change.

Thanks to their framework with freely-evolving atmospheric CO2 concentrations, the authors show that the consideration of the PLA is critical, as it leads to an enhanced greenhouse gas effect in climate forecasts. Indeed it increases by 8 to 20% the global atmospheric CO2 concentrations. This result has great implications for climate forecasts: it highlights the importance of PLA-induced climate changes that a large proportion of current ESM do not consider, and proposes a quantification of this missing part of the atmospheric CO2 content under different climate scenarii (it identifies the PLA-induced changes as a function of climate change itself).

We thank Sarah Berthet for her review and her valuable comments.

However, two points need clarifications from my perspective:
- What implications has the use of an EMIC compared to a classical ESM in terms of feedbacks between the ocean and atmosphere ? This would give some clues on how your main results are generalizable.

Due to the simplified nature of EMICs, they simulate fewer climate feedbacks and simplified dynamics relative to high complexity ESMs. The largest simplification in our setup is the loss of atmospheric dynamics through the use of a 2D EMBM atmosphere, in which the surface wind-stress forcing is prescribed and fixed through time and between simulations. In the "Limitations" section we rephrased by:
"Additionally, if wind stress could evolve freely, as in classical high complexity ESMs, we suppose that the increase in atmospheric temperature would lead to increased wind stress. As a result, upwelling dynamics would be altered."

- One of the main result of this study is that PLA increases the surface net primary production in mid-latitude and upwelling regions due to a higher availability of nutrient concentrations, which is in turn driven by a higher remineralization at the ocean surface (and a reduced export efficiency). While this result is important, I would expect to understand to what perturbations of the oceanic physical conditions the higher remineralization is due.

Phytoplankton light absorption directly increases the surface oceanic temperature; promoting environmental conditions that increase surface remineralization (see equations in Ward et al. (2018)). Consequently, surface nutrient concentrations increase, leading to enhanced surface chlorophyll. Via the phytoplankton light absorption mechanism, the higher chlorophyll allows more heat to be trapped in the ocean surface.

**Specific comments by section**

**Abstract**

**L.9** "This biogeophysical mechanism increases the surface chlorophyll": based on your Table 2, your net primary production (globally integrated) in 2500 increases by less than 2% for all RCPs when activating the PLA. Of course this is not comparable with the order of changes you cited in your introduction part (e.g. "chlorophyll concentration has declined over more than 62% of the ocean surface from 1890 to 2010", "between 1998 and 2006, low surface chlorophyll areas have expanded by 15%"...), but I would highlight that point in the abstract by giving the percentages of changes, because your results show that, by triggering NPP changes of less than 2%, the PLA may perturb the global atmospheric CO2 content by 8 to 20%.

We rephrased to "this biogeophysical mechanism increases the surface chlorophyll by ~2%, the sea surface temperature by 0.2 to 0.6°C, the atmospheric CO2 concentrations by 8-20% and the surface air temperature by 0.3 to 0.9°C".

**L.15** "that may be" or "that are maybe" ?

We rephrased to "may be".

**2 Methods**

From the legend of figure 1, I understood that what differentiates your EMIC from what you call an "ESM of high complexity" here is mainly the use of a simplified atmospheric module ("EMBM") which is not a fully 3D atmospheric model…? Please, could you clarify that aspect in the text of section 2, and explain with one sentence what is EMBM: if not a 3D model, is it a slab layer of atmosphere ?

Yes the atmospheric component, EMBM is not a fully 3D atmospheric model. In the revised manuscript, we revised to:
"The atmospheric component is an Energy Moisture Balance Model (EMBM), which is closely based on the UVic Earth system model (Weaver et al., 2001). It is a 2D slab layer of the atmosphere".

**L.72-74** the authors wrote "EcoGEnIE is an ESM of intermediate complexity (EMIC) (Claussen et al., 2002) and due to the limitations of such a model, we focus on the quantification of the large-scale impacts of phytoplankton light absorption but we do not quantify the components or drivers of those large-scale impacts".
I am wondering how could we trust the large-scale impacts of PLA analyzed here if we do not trust what cause them ? I understand from this sentence that, due to the limitations inherent to an EMIC, the authors do not trust the drivers of the PLA large-scale impacts. Please reformulate.

It is not that we do not trust the drivers, only that we do not attempt to quantify them, which is a layer of complexity deeper than our analysis has gone. The limitation to large-scale impacts is primarily the result of low resolution together with simplified dynamics which cannot address local-scale processes. We have omitted the clause which was causing misunderstanding and have rephrased to:

"EcoGEnIE is an ESM of intermediate complexity (EMIC) (Claussen et al., 2002) and due to the limitations of such a model, in particular its low resolution, we focus on the quantification of the large-scale impacts of phytoplankton light absorption".

**L.75-77** "We chose to conduct our study with an EMIC because we are interested on the effect on particular climate mechanism (e.g. phytoplankton light absorption) and it would have been difficult to isolate this effect with an ESM of high complexity, due to numerous climate feedbacks implemented in high complexity ESM."

In their analysis of many ESM "of high complexity", Séférian et al (2020) decomposed the Earth system interactions represented in ESM involving marine biogeochemistry into 4 main feedbacks: climate-carbon cycle feedbacks (F1), biogenic aerosol-cloud feedbacks (F2), non-CO2 biogeochemical cycle feedbacks (F3) and phytoplankton-light feedbacks (F4). It is not straithforward to me to see how climate feedbacks F1 to F3 would have perturb your analyses of the PLA-induced effects. Please, be more specific : give examples of the numerous feedbacks that would hinder the identification of PLA-induced effects. Don't you mostly think here to ocean-atmosphere interactions (not existing in your case due to the use of a simplified atmosphere with EMBM) ? If true, please mention it.

Séférian, R., Berthet, S., Yool, A. et al. Tracking Improvement in Simulated Marine Biogeochemistry Between CMIP5 and CMIP6. Curr Clim Change Rep 6, 95–119 (2020). https://doi.org/10.1007/s40641-020-00160-0

As stated by the reviewer, we mainly think about ocean-atmosphere interactions. We added the sentences "For instance, in our model setup, the wind is prescribed and doesn't change between simulations. Consequently, the effect of wind on ocean circulation is unchanged between the simulations and through time."

**2.1 Ocean, atmosphere and sea-ice representation**

**L.95** "However, on a global scale, Marsh et al. (2011) show that the model simulates realistic upwelling."

With an horizontal resolution smaller than 3° in latitude (and not specified in longitude... but Ward et al., 2018 declare that they have 10° of longitudinal increments: what about yours ?...) and a minimum vertical spacing of 29 m, I guess "upwelling" refers to equatorial convergence, and not to coastal upwelling regions which have widths < 100 km, associated to very specific coastal dynamics needing quite fine horizontal and vertical resolutions to be represented. But even for equatorial regions, I find a bit inappropriate the expression "the model simulates realistic upwelling" as we know that your model represents only the very large-scale ocean dynamics. Could you describe in the text the dynamical conditions favoring these "realistic" upwellings in your model ?

We used the same horizontal grid as Ward et al. (2018) with a 10° longitudinal increment. Although we did not state that local scale upwelling is realistically modelled - only global scale upwelling - given the reviewer's concerns, and the fact that there are different wind field forcing between Marsh et al. (2011) model setup and our model setup, we have deleted this sentence.

**L.96-107** Again, could you clarify why this 2D atmospheric model was a more suitable choice than the fully 3D atmospheric model PLASIM in your framework ? Could you add a sentence explaining how the use of this simplified atmosphere may help revealing the PLA effects ?

The simplified atmospheric component simplifies climate feedbacks, such as ocean-atmosphere interactions. We refer the reviewer to our answer to her comment l.75-77. We note that the use of the EMBM is well suited for ocean model development given the run time of PLASIM-GOLDSTEIN (50

model years per CPU day) increases the computational demand by several orders of magnitude relative to the EMBM-GOLDSTEIN version (5000 years per model day).

**2.3 Ecosystem community component**

**l.126** "messy feeding" ?

Corrected

**l.145** "so the rate"

Corrected

**2.8 Model inter-comparison**

**l.225** comparison with an other EMIC ("an ESM of intermediate complexity") model: but more generally the reader is curious to know what would give the comparison with a high-complexity ESM?

Thank you for this suggestion. We have now revised Fig 3 to also include a comparison with the CMIP5 ensemble. Independently of the RCP scenario, our increases in SAT are in close agreement with the ensemble mean warming of the EMIC and CMIP5 inter-comparisons, and comfortably within the ensemble ranges.

More generally, I understand that this first comparison focused on surface atmospheric temperature (SAT) because it allows to validate the use of a simplified atmospheric model in this study...? But back to the main goal of this study (effect of the PLA), I would expect here some elements characterizing how the ocean compartment absorbs heat without PLA (ocean heat content or at least ocean temperature). This would allow to discuss later the true PLA effect added by your equation (3). If Zickfeld et al (2013) have no ocean heat data, I suggest you to insert a small paragraph (and figure) characterizing the ocean heat content changes (or time series: see for example Figure 1 of Berthet et al, 2023) for each of your RCPs without and with PLA. Based on eq. (3), the first effect PLA will have on climate before any feedbacks on the biological pump/CHL/atmospheric $CO_2$/SAT, will be to perturb the oceanic temperature, no ? So the first question for me is: how much ocean heat content is altered by the activation of your PLA parameterization ? Could you elaborate a bit on that point ?

Berthet, S., Jouanno, J., Séférian, R., Gehlen, M., and Llovel, W.: How does the phytoplankton–light feedback affect the marine N2O inventory?, Earth Syst. Dynam., 14, 399–412, https://doi.org/10.5194/esd-14-399-2023, 2023.

As requested by the reviewer, we have now computed the time-series of ocean heat content for all the simulations. The ocean heat content anomalies are computed with respect to the mean of the time-series. Independently of the RCP scenario, phytoplankton light absorption increases the ocean heat content in the top 2000m of the water column (see figure below). However, the relative changes in ocean heat content strongly vary between the scenarios. For instance, ocean heat content increases by 79% for RCP2.6 while it only increases by 0.05% for RCP8.5. This is in agreement with our findings, where phytoplankton light absorption has a reduced effect on the climate under RCP8.5 compared to the other RCP scenarios.

We added the figure below in appendix and reported the values of the relative ocean heat content changes in the result section. In the main text, we added the sentences "Phytoplankton light absorption warms the surface of the ocean and increases the ocean heat content (Appendix C1). For

the scenarios RCP2.6, RCP4.5 and RCP6.0, ocean heat content in the top 2000 m increases by 16-79%. For RCP8.5, this increase is strongly reduced to 0.05% reflecting the lower increase in global surface chlorophyll under this scenario.".

[Figure]

**3 Results**

**3.1 Oceanic properties**

**l.240** I would suggest to reformulate: "when PLA is activated" or "represented", rather than "simulated" ; this is only a suggestion, as my english is far from being perfect.

We changed to "activated".

**L.241** How is the spatial pattern of this POC flux reduction ? Do you observe a reduction over the entire globe ? Or is it consistent with the patterns you described for the chlorophyll (l.266-270), i.e. mainly marked in upwelling and mid-latitude regions ?

We plotted the difference of surface POC flux between the simulations (see below). The maps represent the simulation without minus with phytoplankton light absorption. Red colours represent a higher POC flux for the simulations without phytoplankton light absorption while blue colours represent a lower POC flux without phytoplankton light absorption compared to the corresponding simulation with this biogeophysical mechanism.
The patterns of POC flux reduction is consistent with the patterns described for chlorophyll. The reduced surface POC flux is mainly marked in the upwelling and mid-latitude regions.

[Figure]

**L.242** "independently of the RCP scenario": by activating the PLA, the oceanic temperature increases in sub-surface in all scenarii, but with different intensities, no ?

Phytoplankton light absorption leads to an increase in temperature along the whole water column (Figure C1 in Appendix and the figure about ocean heat content above).

**l.245** "Indeed, the surface net primary production increases with phytoplankton light absorption": could you explain why ? To what perturbations of the physical conditions is it due (see my general comment on section 2.8) ? Please elaborate on that.

Phytoplankton light absorption increases oceanic temperature, promoting environmental conditions that increase surface remineralization (Ward et al., 2018), increasing nutrient concentration and thus surface net primary production. We added the sentence:
"Indeed, phytoplankton light absorption increases the oceanic temperature (see below), promoting environment conditions that increase the surface remineralization (Table 2). Consequently, the nutrient concentration (Table F1) and thus the net primary production in the surface ocean layer is enhanced."

**3.1.2 Surface chlorophyll**

**L.266** Unlike the results of Paulsen (2018), who reports a decline in chlorophyll concentrations in the upwelling regions with PLA (L.46), you find a higher chlorophyll (CHL) concentration in the upwelling and mid-latitude regions with PLA in your model and framework: could you explain why Paulsen obtained an opposite feedback with its "Earth system model of high complexity" ? By which mechanism ? Does your EMIC represent this mechanism ? Or is this different behaviour attributable to the fact that Paulsen run its ESM under prescribed future atmospheric CO2 concentrations rather than freely-evolving emissions: in this case could you explain by which mechanism the atmospheric CO2 concentration may constrain the CHL to decrease locally in upwelling regions ?

Paulsen et al. (2018) reports a decline in chlorophyll in the upwelling regions with PLA because her ESM simulates a decline in the strength of the upwelling (see section 4.5.3 of Paulsen et al. (2018)). The reduced upwelling strength is caused by a shoaling of the oceanic mixed layer depth and a decrease of the trade winds (Paulsen et al., 2018). In contrast, our results show an increase in chlorophyll in upwelling regions because, in these regions, the upward vertical velocity is enhanced by phytoplankton light absorption (increased upwelling strength).

We apologize for the confusion here because in the previous version of the manuscript we wrote that "Paulsen et al. (2018) reports an enhanced circulation in the upwelling regions", which is not true. We rephrased by:

"With phytoplankton light absorption, Paulsen et al. (2018) reports a decline in chlorophyll concentrations in the upwelling regions due to a reduced upwelling strength. However, this local effect is outweighed by advective process such as the upward transport of warmer subsurface water originating off-equator, leading a local oceanic warming of up to 0.7°C."

**3.1.3 Sea surface temperature**

**L.273** "Due to changes in surface chlorophyll, we expect variations in SST".
What do you mean exactly here ? Due to changes in 1) surface chlorophyll concentration or 2) in absorption properties of surface chlorophyll ? This does not imply the same chain of causality:
– case 1) describes the fact that PLA activation directly affects CHL concentration and, then, indirectly affects the SST due to the CHL concentration changes. However, in this case, could you clarify what mechanism triggers the initial perturbation of your CHL concentration ? In other words, how the PLA activation affects your CHL concentration ?
– case 2) describes the fact that activating PLA has first a direct effect on ocean temperature. And that the other effects on CHL/export/remineralization arise from that one.

We meant that we expect variations in SST because activating phytoplankton light absorption has a direct effect on oceanic temperature (case 2). To avoid confusion, we removed this sentence.

**3.2.1 Atmospheric CO2 concentration**

**L.293** I am a bit puzzled about these runs driven by CO2-emissions that do not match the target. I am not sure to fully understand the implications that could have on your analyses. Could you elaborate on it ?

Generating a tuned version of a new climate-carbon-cycle model coupling is highly demanding. We chose to tune the model's primary production to generate realistic plankton biomass and export production. We focus on the relative changes between simulations with and without phytoplankton light absorption and so are less interested in the absolute climate state. Although a differently tuned model would (by definition) exhibit differences in any state-dependent feedbacks, we demonstrate in Figure 3 that the carbon cycle response is largely insensitive to this background state, and the model reflects the responses to a wide range of emissions shown by EMIC and CMIP5 ensembles. We rephrased:

"We are interested in relative changes between our simulations rather than the absolute values, so the specifics of the background state are unlikely to affect the qualitative findings of the study, especially given that the carbon-cycle response to a range of emissions is consistent with IPCC model intercomparison projects (Figure 3)."

**L.302** "For the RCP8.5 scenario, the atmospheric CO2 concentration increases by 8% only, which is due to the lower increase in chlorophyll and SST".

To demonstrate this assertion it would be interesting to see maps of atmospheric and oceanic CO2 partial pressure, as well as DpCO2 for all RCPs. Because in the current state I am not sure you have enough elements to directly conclude what you wrote. For me, you need to disentangle here 1) how the PLA activation changes your ocean CO2 content, from 2) how the CO2 atmospheric content in RCP8.5 allows to absorb new oceanic outgassing compared to the other RCPs. Your results in figures 7 and 8 show a non-linear behaviour of RCP8.5 compared to the other three RCPs, which is most likely attributable to a non-linear effect of the increased atmospheric CO2 concentrations.

Due to the 2D atmospheric component of our model, the atmospheric pCO2 is uniform over the whole world. Rather than showing maps, we report here and in appendix the changes in oceanic CO2 concentration and in atmosphere-ocean CO2 fluxes. The table below summarizes these results; negative values in the second column indicate a lower oceanic CO2 concentration in the simulations with phytoplankton light absorption while positive values in the third column indicate a higher outgassing of CO2 with phytoplankton light absorption. The decrease in oceanic CO2 concentration and the increase in air-sea CO2 flux are reduced under RCP8.5 scenario compared to the other scenarios. This result shows, again, the non-linear behaviour of RCP8.5 compared to the other three RCPs. We rephrased by:
"For the RCP8.5 scenario, the atmospheric CO2 concentration increases by only 8%. This lower increase in atmospheric CO2 concentration is due to the lower decrease in oceanic CO2 content and lower increase in air-sea CO2 flux (Appendix D1). These reduced changes in between the different CO2 reservoirs are due to the lower increase in chlorophyll and SST."

|  | Oceanic CO2 concentration | Air-sea CO2 flux |
|---|---|---|
| RCP2.6 | -14.0% | +17.4% |
| RCP4.5 | -12.1% | +15.6% |
| RCP6.0 | -11.4% | +14.8% |
| RCP8.5 | -2.0% | +1.3% |

From my perspective you did not dig deep enough into this aspect, because it was one of the main conclusions of your work: the effect of PLA activation is not linear and depends on the climate scenario. You may show that RCP8.5 crossed a tipping point (due to an extensive ice melt and other effects) what changes the way the ocean-atmosphere system manages the CO2 exchanges and finally, modulates the effect/amplitude the PLA activation may have on climate.

In the general discussion section we added the sentence:
"Our results show that there is the potential of a tipping point (which is crossed in RCP8.5 in our model) at which changes in the climate system modulate the effect of phytoplankton light absorption."

**4.1 General discussion**

**L.334** "Our results show that phytoplankton light absorption affects water temperature and nutrient concentrations."
Please see my related comment in section 3.1.3.

We rephrased by:
"Our results show that phytoplankton light absorption has a direct effect on oceanic temperature, affecting, in consequences, the biogeochemical properties of the ocean."

**L.335** "The increase in surface nutrient concentrations (Appendix D1) is driven by a reduced export efficiency of organic matter and enhanced remineralization at the ocean surface (Table 2)."

While I found the result of more remineralization in surface very interesting, I still have the feeling that something is missing in your analyses. You did not explain (or I missed it, so maybe it would be great to clarify it) by what mecanisms does the PLA activation affect your modelled remineralization and export ? This will possibly also help understand why RCP8.5 does not react proportionally to the other RCPs when activating the PLA.

The modelled remineralization and export depend on the oceanic temperature (Ward et al., 2018). The activation of phytoplankton light absorption increases oceanic temperature, increasing in turn the surface remineralization and reducing the POC export flux. We rephrased to:
"Activating phytoplankton light absorption directly increases the oceanic temperature, which reduces export efficiency of organic matter and enhanced remineralization at the ocean surface. Consequently, the surface nutrient concentration increases, which leads to a higher chlorophyll. Via the phytoplankton light absorption mechanism, this higher chlorophyll creates a feedback loop, leading to a warmer ocean."

**L.336** "The increased surface nutrient concentrations leads to higher surface chlorophyll, which in turn leads to a warming of the ocean surface."
Here I interpreted that you choose the case (1) of my comment in section 3.1.3.

Phytoplankton light absorption has a direct effect on oceanic temperature, which in turn affects the biogeochemical properties of the ocean (case 2). See answer to comment L.335 (just above).

**4.2 Limitations**

**L. 384** "Our results highlight that phytoplankton light absorption itself increases chlorophyll leading to more heat being trapped in the ocean surface."
My guess is that PLA promotes environmental conditions in ocean surface temperature that allow an increase in remineralization in surface, what triggers an increase in nutrients concentrations and, thus, an increase in CHL concentrations allowing more heat to be trapped: is that what you mean ? Please clarify

We agree with the reviewer and rephrased by:
"Our results highlight that phytoplankton light absorption increases ocean surface temperature, allowing an increase in surface remineralization and triggering an increase in nutrient concentrations. As a consequence, chlorophyll increases allowing more heat to be trapped in the ocean surface."